# Indispensable role of Galectin-3 in promoting quiescence of hematopoietic stem cells

Weizhen Jia[1], Lingyu Kong[2], Hiroyasu Kidoya [1], Hisamichi Naito [1], Fumitaka Muramatsu[1], Yumiko Hayashi[1], Han-Yun Hsieh[1], Daishi Yamakawa[3], Daniel K. Hsu[4], Fu-Tong Liu[4,5] & Nobuyuki Takakura [1✉]

Hematopoietic stem cells (HSCs) in adult bone marrow (BM) are usually maintained in a state of quiescence. The cellular mechanism coordinating the balance between HSC quiescence and differentiation is not fully understood. Here, we report that galactose-binding lectin-3 (galectin-3; Gal-3) is upregulated by Tie2 or Mpl activation to maintain quiescence. Conditional overexpression of Gal-3 in mouse HSCs under the transcriptional control of *Tie2* or *Vav1* promoters (Gal-3 Tg) causes cell cycle retardation via induction of p21. Conversely, the cell cycle of long-term repopulating HSCs (LT-HSCs) in Gal-3-deficient (Gal-3$^{-/-}$) mice is accelerated, resulting in their exhaustion. Mechanistically, Gal-3 regulates *p21* transcription by forming a complex with Sp1, thus blocking cell cycle entry. These results demonstrate that Gal-3 is a negative regulator of cell-cycling in HSCs and plays a crucial role in adult hematopoiesis to prevent HSC exhaustion.

[1] Department of Signal Transduction, Research Institute for Microbial Diseases, Osaka University, Suita, Osaka, Japan. [2] Department of Head and Neck Surgery, Harbin Medical University Cancer Hospital, Harbin, Heilongjiang, China. [3] Department of Physiology, Mie University Graduate School of Medicine, Tsu, Mie, Japan. [4] Department of Dermatology, School of Medicine, University of California-Davis, Sacramento, CA, USA. [5]Present address: Institute of Biomedical Sciences, Academia Sinica, Taipei, Taiwan. ✉email: ntakaku@biken.osaka-u.ac.jp

Ensuring the survival and proliferation of hematopoietic stem cells (HSCs) in long-term culture is required to facilitate genetic modification prior to transplanting these cells to treat hematopoietic diseases in patients with genetic disorders[1–3]. However, maintenance of the HSCs in an undifferentiated state in vitro remains a major obstacle. To overcome this hurdle, characterization of the stem-cell niche is important for understanding the molecular mechanisms whereby stem cells differentiate or self-renew. Moreover, identification of the niche cells would help to identify the molecular cues that regulate "stemness"[4,5].

Recent studies have shown each of the multiple types of niche cells, i.e., osteoblasts[6], endothelial cells (ECs)[7], C-X-C motif chemokine ligand 12 (CXCL12)-abundant reticular (CAR) cells[8], mesenchymal stem cells[9], arteriolar pericytes[10], and sympathetic nerves[11], is important for regulating the balance between HSC quiescence and differentiation. Most studies have shown that extrinsic signals from niche cells affect HSCs by influencing cell cycle-related molecules in HSCs. For example, extrinsic factors including Notch ligands, sonic hedgehog, Wnt3a and others regulate the expression and function of Bmi1, PTEN, p21, and p57[12–18]. Of the many cytokine receptors expressed by HSCs, Tie2 and Mpl have been reported to be involved in maintaining the quiescent state. Although it has been argued that Angiopoietin-1 (Ang-1)/Tie2 has a nonessential role, enhanced signaling via Tie2 by Ang-1, and Mpl signaling by Thrombopoietin (Thpo), seem to induce HSC quiescence by activating cell adhesion molecules such as $\beta$1-integrin and N-cadherin[19,20]. However, the mechanisms by which Tie2 and Mpl maintain quiescence have not been fully elucidated.

Galectins are a family of animal lectins with conserved carbohydrate-recognition domains (CRDs) for beta-galactoside[21]. So far, 15 mammalian Galectins (Gal-1-15) have been identified, all containing a conserved CRD consisting of approximately 130 amino acids. Based on the number and the organization of CRDs, members of the Galectin family have been classified into three subtypes: the prototype group, the tandem repeat group, and the chimera group. Members of the prototype group (Gal-1, -2, -5, -7, -10, -11, -13, -14, and -15) contain one CRD. Gal-4, -6, -8, -9, and -12, the members of the tandem repeat group are composed of a single polypeptide chain that forms two distinct but homologous CRDs, separated by an unconserved linker sequence of up to 70 amino acids. Gal-3, the only vertebrate chimera-type galectin, also contains one CRD which is connected to an unusually long N-terminal proline- and glycine-rich domain, responsible for multimer formation[22,23]. Multiple functions of Gal-3 have been reported depending on its location. Extracellular Gal-3 is secreted via a non-classical pathway (because of lack of signal peptides) and can bind to the cell surface through glycosylated proteins, thereby triggering or modulating cellular responses such as mediator release or apoptosis. Intracellular Gal-3 has been reported to inhibit apoptosis, regulate the cell cycle, and participate in the nuclear splicing of pre-mRNA. Through specifically interacting with a variety of intra- and extracellular proteins, Gal-3 affects numerous biological processes and seems to be involved in different physiological and pathophysiological conditions, such as development, immune reactions, and tumorigenesis[23,24]. However, the function of Gal-3 in the HSC system has not been well studied.

In this work, we determine that Gal-3 is highly expressed in long-term repopulating HSCs (LT-HSCs) localizing in the vascular niche. By using genetically manipulated mice, i.e., gene ablation and overexpression, we analyze the roles of Gal-3 in quiescent HSCs. We also elucidate the mechanisms regulating Gal-3 induction in association with Tie2 or Mpl activation and its functional relevance for maintaining quiescence in HSCs through the regulation of *p21* transcription.

## Results

**Gal-3 is highly expressed in quiescent LT-HSCs.** Previous research suggested that Gal-3 expressed in macrophages participates in regulating the inflammatory response[25,26]. However, the role of Gal-3 and the mechanisms regulating its expression in immature hematopoietic cells including HSCs are still not clear. Therefore, we quantified *Gal-3* mRNA and protein levels in LT-HSCs [CD150$^+$CD48$^-$Flt3$^-$ cells within the Lineage$^-$Sca-1$^+$c-Kit$^+$ (LSK) population], short-term repopulating HSCs (ST-HSCs; CD150$^-$CD48$^-$Flt3$^-$ cells within the LSK population) and multipotent progenitors (MPPs) within the LSK population, such as MPP2 (CD150$^+$CD48$^+$Flt3$^-$), MPP3 (CD150$^-$CD48$^+$Flt3$^-$), and MPP4 (CD150$^-$CD48$^+$Flt3$^+$). We found that *Gal-3* is highly expressed in LT-HSCs and at lower levels in ST-HSCs and MPPs2-4 (Fig. 1a). Consistent with this, we also found that side-population (SP)$^{low}$ cells and CD34$^-$Flt3$^-$ cells within LSK cells (both of which are LT-HSC enriched populations[19,27]) expressed more Gal-3 than SP$^{high}$ LSK cells, CD34$^+$Flt3$^-$ LSK cells or CD34$^+$Flt3$^+$ LSK cells (Supplementary Fig. 1a, b). Staining cells from the hematopoietic fraction with anti-Gal-3 antibody (Fig. 1b), revealed that this protein was more frequently present in LT-HSCs than ST-HSCs or MPPs2-4 (Fig. 1c). Particularly, the nuclear localization of Gal-3 was especially clear in LT-HSCs.

Appropriate interactions of HSCs with their specific microenvironment (niche) is crucial for maintaining stem-cell properties. Accumulating evidence indicates that here, the vascular niche plays a critical role in HSC maintenance[7,10]. Thus, we examined the distribution of Gal-3$^+$ progenitors including HSCs by c-Kit positivity in adult bone marrow (BM) and found that Gal-3$^+$c-Kit$^+$ cells are located close to endomucin$^+$ ECs (Supplementary Fig. 1c). Similarly, we observed that Gal-3$^+$ CD150$^+$ cells were localized close to ECs (Fig. 1d, e).

Next, we induced BM suppression using 5-fluorouracil (5-FU) which causes apoptosis of actively cycling cells, in order to determine whether Gal-3 expression is enriched in quiescent HSCs (as established by c-Kit and CD150 expression). Immunofluorescence staining revealed that Gal-3 is highly expressed by the residual CD150$^+$ or c-Kit$^+$ cells 2 days after 5-FU injection (Supplementary Fig. 1d, e). Simultaneously, the proportion of LT-HSC in the G$_0$ phase decreased, suggesting that hematopoietic reconstitution had started (Supplementary Fig. 1f). Four days after 5-FU injection, the lowest amount of LT-HSC in G$_0$ phase during the observation period was recorded (Supplementary Fig. 1f) and the *Gal-3* mRNA level was significantly decreased in these cells (Supplementary Fig. 1e). Gal-3 protein positivity in CD150$^+$ or c-Kit$^+$ cell was also reduced on day 4 (Supplementary Fig. 1d). Moreover, from day 6 after 5-FU injection, the proportion of LT-HSC in G$_0$ phase gradually increased, and *Gal-3* mRNA and protein levels gradually recovered (Supplementary Fig. 1d–f). These results suggest that changes of Gal-3 expression in LT-HSC play a role in controlling HSC quiescence during BM reconstitution. In addition, we also analyzed the spatial relationship between Gal-3$^+$ HSCs and vascular structures after 5-FU injection. A dilation of the BM vascular lumen was observed after the injection of 5-FU and HSCs detached from vascular ECs and changed their morphology into spindle-like shapes (Fig. 1f, Supplementary Fig. 1d). We found that the percentage of Gal-3$^+$ HSCs localizing adjacent to the vasculature (within 15 $\mu$m) 8 days after 5-FU injection was significantly increased compared with this value 4 days after (63% and 43%, respectively). However, this was not seen with Gal-3$^-$ HSC (Fig. 1g). These results therefore suggest that Gal-3$^+$ HSCs interact with cells in the vascular niche and control the cell cycle of LT-HSCs during the process of BM reconstitution.

**Loss of Gal-3 induces differentiation of LT-HSCs.** To investigate the physiological functions of Gal-3 in HSCs, we used Gal-3-

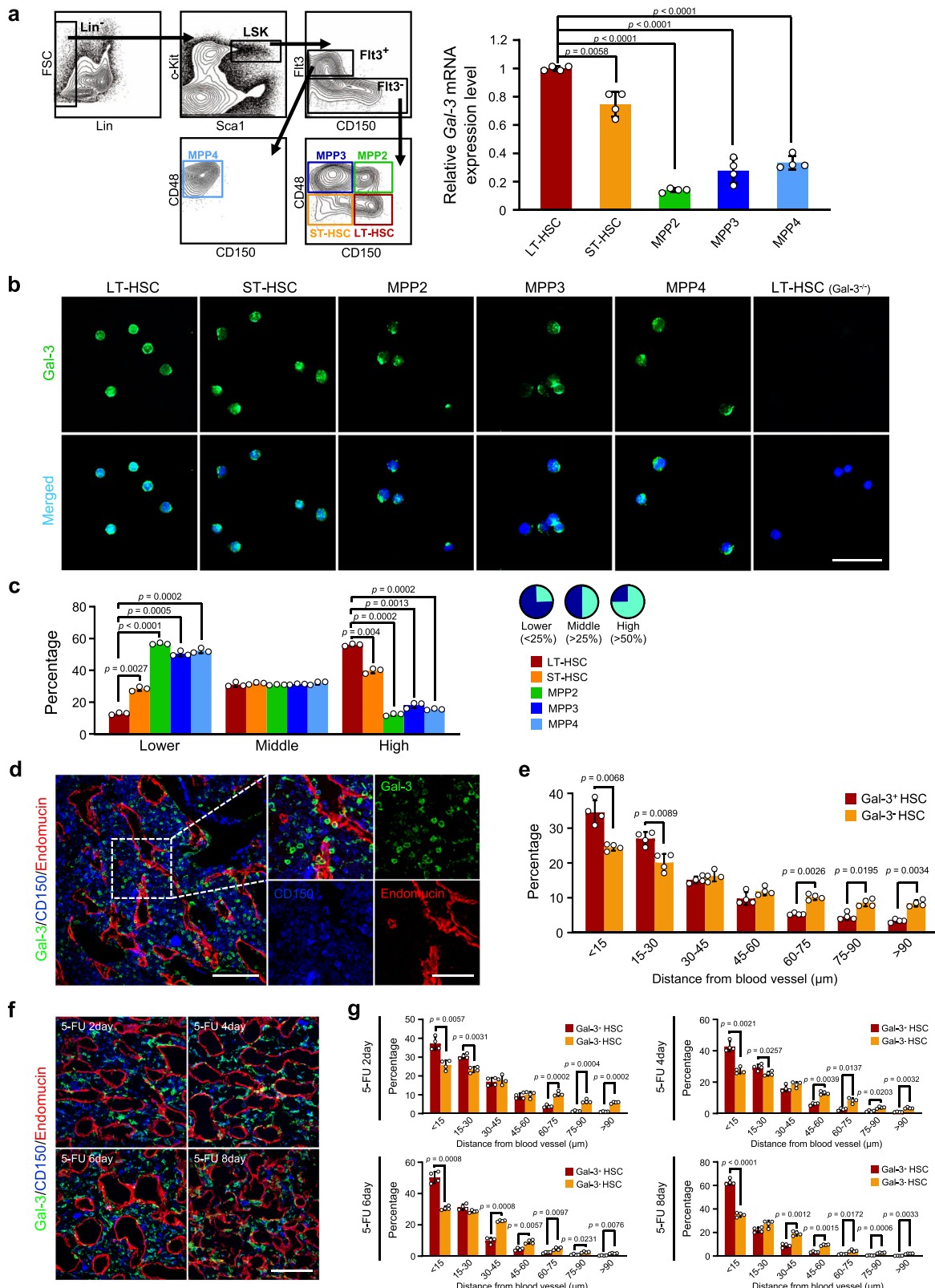

deficient mice (Gal-3$^{-/-}$) which are viable, fertile and do not exhibit any overt defects. However, these mice had lower responses to inflammatory stimuli in the peritoneal cavity, mainly due to the reduction of infiltrating macrophages[28]. Moreover, it has been reported that the number of peripheral leukocytes is increased in adult Gal-3$^{-/-}$ mice and this may reflect abnormal differentiation of HSCs[29]. As there was no information on the

integrity of HSCs regulated by Gal-3, we first analyzed LSK populations of HSCs in the BM from wild-type (WT; Gal-3$^{+/+}$) or Gal-3$^{-/-}$ mice. Compared with age-matched Gal-3$^{+/+}$ mice, Gal-3$^{-/-}$ mice had both higher percentages and absolute numbers of LSK cells (Fig. 2a, b). Moreover, we assessed CD150, CD48, and Flt3 expression on the LSK cell population to quantify LT-HSCs, ST-HSCs, and MPPs2-4. The results indicated that

**Fig. 1 Gal-3 is highly expressed by LT-HSCs in the vascular niche. a** Relative expression level of *Gal-3* mRNA in LT-HSC, ST-HSC, MPP2, MPP3, and MPP4 cells in the BM of WT mice (LT-HSC: CD150$^+$CD48$^-$Flt3$^-$LSK; ST-HSC: CD150$^-$CD48$^-$Flt3$^-$LSK; MPP2: CD150$^+$CD48$^+$Flt3$^-$LSK; MPP3: CD150$^-$CD48$^+$Flt3$^-$LSK; MPP4: CD150$^-$CD48$^+$Flt3$^+$LSK). $n = 4$ biological replicates, compared by two-sided *t* test. Data are presented as mean values ± S.D. **b** Detection of Gal-3 protein (green) by immunofluorescence (IF) in LT-HSC, ST-HSC, MPP2, MPP3, and MPP4 populations in the BM of WT mice. DAPI (blue) was used to detect nuclei. A representative image is shown ($n = 3$ independent experiments with at 2–3 cell section per experiment). Scale bar, 50 μm. **c** Quantification of the IF analysis developed as described in **b**. For quantification, three cell sections (per each cell type) were analyzed (ten random fields per section) using Image J software (two-sided *t* test). Data are presented as mean values ± S.D.. Right-hand panels show the ratio between Gal-3 (green) and nuclear area (blue) which was calculated to assess the diversity of Gal-3 expression. **d** IF staining of Gal-3 (green), CD150 (blue), and Endomucin (red) in BM sections from WT mice. Endomucin was selected as an endothelial cell marker in BM and the dashed box indicates areas shown at higher magnification. A representative image is shown ($n = 4$ biological replicates). Scale bars, 100 μm and 50 μm (inset). **e** Distances between Gal-3$^+$ HSC (Gal-3$^+$CD150$^+$) or Gal-3$^-$ HSC (Gal-3$^-$CD150$^+$) and blood vessels (Endomucin$^+$) in the metaphysis and diaphysis area of WT BM. $n = 4$ biological replicates (six random fields per BM sections), compared by two-sided *t* test. Data are presented as mean values ± S.D. **f** IF staining of Gal-3 (green), CD150 (blue), and Endomucin (red) in BM sections from WT mice at different time points after 5-FU treatment. Endomucin was selected as above. A representative image is shown ($n = 4$ biological replicates). Scale bar, 100 μm. **g** Distances between Gal-3$^+$ HSC (Gal3$^+$CD150$^+$) or Gal-3$^-$ HSC (Gal3$^-$CD150$^+$) and blood vessels (Endomucin$^+$) in the metaphysis and diaphysis area of WT BM at different time points after 5-FU treatment. $n = 4$ biological replicates at each time point (six random fields per BM sections), compared by two-sided *t* test. Data are presented as mean values ± S.D.

both percentages and absolute numbers of LT-HSCs and MPP2 decreased, but ST-HSCs, MPP3, and MPP4 increased in Gal-3$^{-/-}$ mice (Fig. 2c, d). These data suggest that Gal-3 is involved in the maintenance of the stem-cell pool in LT-HSCs, which are otherwise prone to differentiate into progenitors in the absence of Gal-3.

In order to verify that we were detecting the stem-cell population, we also employed other markers of these cells. We labeled the LT-HSCs with Hoechst 33342 dye to analyze the SP cell population. This revealed that both the percentages and absolute numbers of SP$^{low}$ LSK cells were lower in the BM of Gal-3$^{-/-}$ relative to Gal-3$^{+/+}$ mice (Supplementary Fig. 2a, b). Furthermore, we analyzed the LT-HSC frequencies and absolute counts within the CD34$^-$Flt3$^-$LSK fractions. The percentage and absolute number of LT-HSCs was significantly decreased in BM of Gal-3$^{-/-}$ mice relative to their Gal-3$^{+/+}$ counterparts (Supplementary Fig. 2c, d). These data are consistent with the results obtained using CD150, CD48, and Flt3 as markers of HSCs. In addition, we determined how the ability of HSCs to differentiate into the myeloid and lymphoid lineages was affected by the lack of Gal-3. Although it has been reported that leukocyte proliferation is induced in Gal-3$^{-/-}$ mice[29], we found no significant differences in the numbers of lineage-committed cells in Gal-3$^{+/+}$ and Gal-3$^{-/-}$ mice (Supplementary Fig. 2e, f). To confirm the Gal-3-independent manner in which differentiation was occurring, we used colony-forming assays [i.e., cobblestone area-forming cell (CAFC) and colony-forming units in culture (CFU-C) assays] to assess whether Gal-3 deficiency affects HSC differentiation. We found that after in vitro culture for 1 week, the number of colonies formed by BM mononuclear cells (BMMNCs) from Gal-3$^{-/-}$ mice in the CAFC assay was significantly higher than from Gal-3$^{+/+}$ mice (Fig. 2e). In the CFU-C assay, the absolute number of colonies, especially CFU-GEMM colonies, was significantly higher in LSK cells from Gal-3$^{-/-}$ relative to Gal-3$^{+/+}$ BM (Fig. 2f), but without any significant differences in the proportions of each cell type (Fig. 2g).

As noted above, we found that Gal-3$^{-/-}$ LSK cells formed more GEMM clones, suggesting that stem cells are susceptible to differentiation into progenitors. Thus, we further analyzed the ability of HSCs to proliferate and differentiate in vivo using CFU-in spleen (S) assays. We found that many colonies were formed in spleen when Gal-3$^{-/-}$ BM cells were transplanted 8 days after BM-transplantation (BM-T); however, at 13 days after BM-T, the number of colonies was sharply decreased and was significantly lower than when Gal-3$^{+/+}$ BM cells were transplanted (Fig. 2h). This suggests that Gal-3$^{-/-}$ BM-derived HSCs possess reduced long-term self-renewal ability. To confirm this hypothesis, we

performed an in vitro long-term culture-initiating cell (LTC-IC) assay and found that Gal-3 deficiency significantly decreased the number of LTC-ICs relative to those from Gal-3$^{+/+}$ animals (Fig. 2i). These results suggest that Gal-3 deficiency causes accelerated HSC differentiation.

To eliminate the possibility that Gal-3 deficiency in the stromal cell component of the BM, rather than in the HSCs themselves, caused the accelerated differentiation phenotype. We established a BM-T chimeric mouse model by transplanting WT BM-derived LSK cells into lethally-irradiated Gal-3$^{+/+}$ or Gal-3$^{-/-}$ mice (Supplementary Fig. 3a). Sixteen weeks after BM-T, we found that body weights and spleen weights did not differ significantly between the two groups of recipients (Supplementary Fig. 3b, c). Next, we analyzed the frequencies and absolute numbers of LSK cells in the spleen and BM from these animals. Chimeras exhibited no differences between Gal-3$^{+/+}$ and Gal-3$^{-/-}$ recipients in the frequencies and absolute numbers of LSK cells in the spleen and BM (Supplementary Fig. 3d–i). In addition, we also found that the absolute numbers and frequency of LT-HSCs were not significantly different between Gal-3$^{+/+}$ and Gal-3$^{-/-}$ recipients (Supplementary Fig. 3j–l). Last, to test BM reconstitution in the BM-T chimeric mouse model, we injected 5-FU into these animals as well. This revealed that Gal-3 deletion in BM niche cells did not affect HSC reconstitution, and that the frequency and absolute numbers of donor-derived HSCs were similar in Gal-3$^{+/+}$ and Gal-3$^{-/-}$ recipients (Supplementary Fig. 3m–o). These results therefore suggest that HSC dysfunction in Gal-3$^{-/-}$ mice is intrinsic to the HSCs, and that Gal-3 deficiency in the BM microenvironment (niche cells) does not affect HSC function.

**Gal-3 deficiency accelerates LT-HSC cell cycle.** Next, we examined the function of Gal-3 in LT-HSCs. Previous studies had indicated that Gal-3 forms heterodimers with Bax, mediated by the NWGR motif located in the CRD of Gal-3. This results in antiapoptotic activity in several cancers, thus counteracting the effects of some chemotherapeutic drugs[30]. Based on these studies, we asked whether Gal-3 deficiency increases apoptosis in LT-HSCs, but we found that neither percentages nor absolute numbers of apoptotic cells in the LT-HSCs were different in Gal-3$^{+/+}$ and Gal-3$^{-/-}$ mice (Supplementary Fig. 4a, b).

It has been reported that another function of Gal-3 is to regulate the cell cycle, for example, its ability to influence cell-cycle arrest in human breast epithelial cells[31]. Therefore, we next investigated the cell cycle status of LT-HSCs and found that the percentage of LT-HSCs in G$_0$ was decreased in BM from Gal-3$^{-/-}$ relative to Gal-3$^{+/+}$ mice. Reciprocally, increased percentages of cells in G$_1$

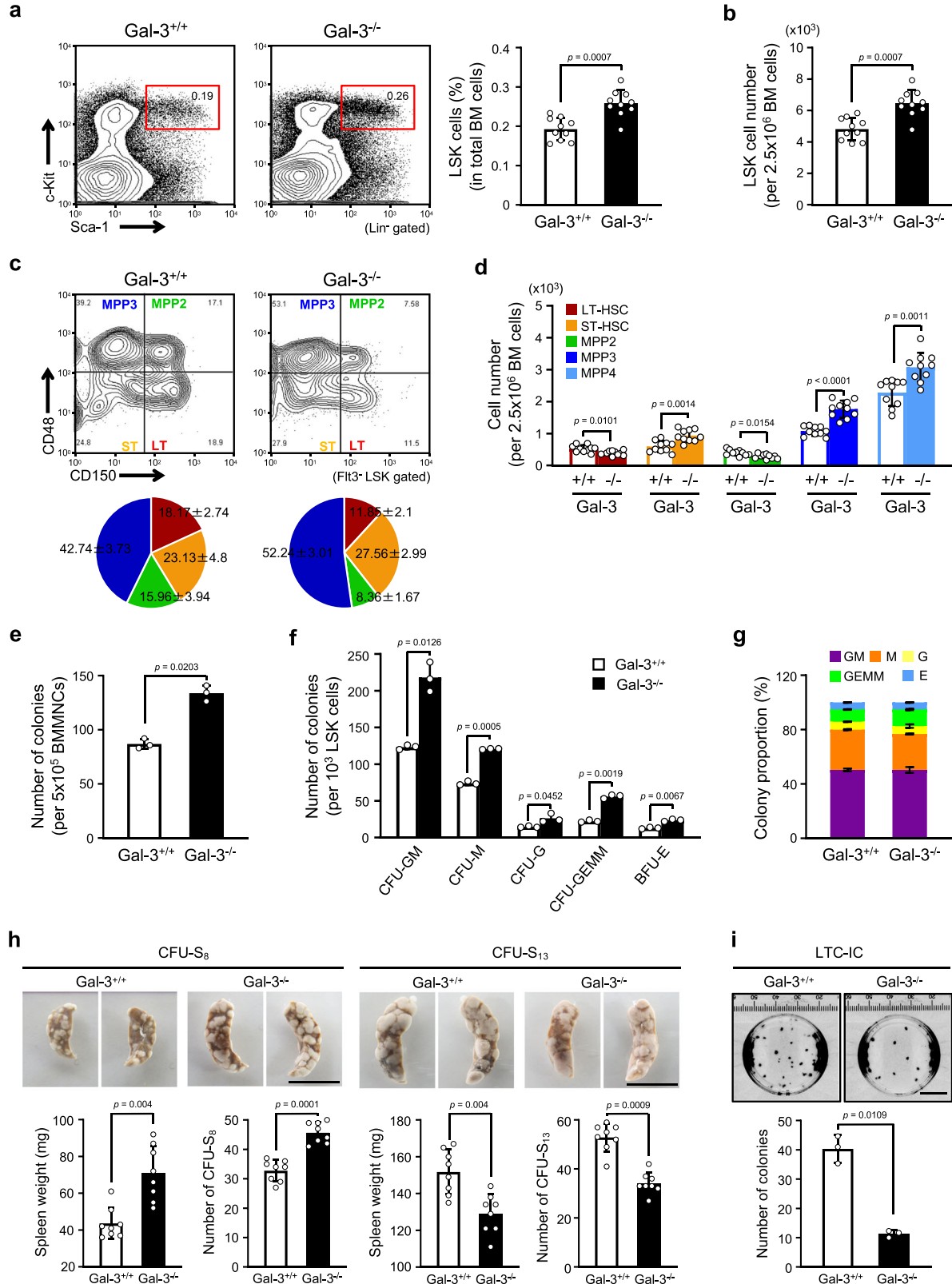

and $S/G_2/M$ were seen in BM from Gal-3$^{-/-}$ mice (Fig. 3a). In order to compare the proliferation of BM LT-HSC from Gal-3$^{+/+}$ and Gal-3$^{-/-}$ mice, we performed an in vivo EdU incorporation assay. Fig. 3b shows a higher rate of EdU incorporation by the LT-HSC of Gal-3$^{-/-}$ than Gal-3$^{+/+}$ mice (Fig. 3b).

In addition, we used an in vitro model to study the influence of Gal-3 on the cell cycle in the pro-B cell line Ba/F3. We

overexpressed Gal-3 protein in these cells (BaF/Gal-3 #1, #2) (Supplementary Fig. 4c) and compared their proliferation with BaF/mock cells (control). We found that proliferation of BaF/Gal-3 cells was significantly decreased (Supplementary Fig. 4d). We also cultured both BaF/mock and BaF/Gal-3 cells in the absence of serum for 12 h, and then with serum and IL-3 again for 24 h after the 12 h starvation. As shown in Supplementary Fig. 4e, Gal-

**Fig. 2 Loss of Gal-3 induces proliferation of LT-HSCs in the adult BM. a** (Left) Representative flow cytometric analysis comparing Gal-3[+/+] with Gal-3[−/−] LSK cell frequencies (red box). (Right) Bar graph showing the percentage of LSK cells among total BM cells ($n = 10$ biological replicates per genotype, compared by two-sided $t$ test). Data are presented as mean values ± S.D. **b** Absolute number of LSK cells in BM of Gal-3[+/+] or Gal-3[−/−] mice ($n = 10$ biological replicates per genotype, compared by two-sided $t$ test). Data are presented as mean values ± S.D. **c** (Top) Representative flow cytometric analysis of LT-HSC, ST-HSC, MPP2, and MPP3 populations in BM of Gal-3[+/+] or Gal-3[−/−] mice. (Bottom) Pie charts represent average percentages of the four subsets within the Flt3−LSK population ($n = 10$ biological replicates per genotype, compared by two-sided $t$ test). Data are presented as mean values ± S.D. and exact $p$ values: LT-HSC: $p = 0.00022$ vs. Gal-3[−/−]; ST-HSC: $p = 0.0107$ vs. Gal-3[−/−]; MPP2: $p = 0.00017$ vs. Gal-3[−/−]; MPP3: $p = 0.00073$ vs. Gal-3[−/−]. **d** Absolute number of LT-HSCs, ST-HSCs, MPP2, MPP3, and MPP4 in BM of Gal-3[+/+] or Gal-3[−/−] mice ($n = 10$ biological replicates per genotype, compared by two-sided $t$ test). Data are presented as mean values ± S.D. **e** Number of colonies formed from Gal-3[+/+] or Gal-3[−/−] BMMNCs in CAFC assays ($n = 3$ biological replicates per genotype, compared by two-sided $t$ test). Data are presented as mean values ± S.D. **f** Hematopoietic lineage development was assessed by CFU-C assay. Purified LSK cells in BM of Gal-3[+/+] or Gal-3[−/−] mice and cultured in MethoCult GF M3434 media (CFU-GM CFU-granulocyte/macrophage, CFU-M CFU-macrophage, CFU-G CFU-granulocyte, CFU-GEMM CFU-multipotential progenitor, BFU-E burst-forming unit-erythroid). $n = 3$ biological replicates per genotype, compared by two-sided $t$ test. Data are presented as mean values ± S.D. **g** Percentages of different colony types developed as described in **f** ($n = 3$ biological replicates per genotype). **h** CFU-S assay. Representative images of spleens isolated from recipient mice 8 (top left) or 13 (top right) days after transplantation of Gal-3[+/+] and Gal-3[−/−] BM cells. Quantification of splenic weight and colonies in recipient mice 8 (bottom left) or 13 (bottom right) days after transplantation ($n = 8$ biological replicates per genotype, compared by two-sided $t$ test). Data are presented as mean values ± S.D. Scale bars, 1 cm. **i** Quantification of long-term reconstituting HSCs by LTC-IC assay showing images of colonies (top) and total colony numbers (bottom). $n = 3$ biological replicates per genotype, compared by two-sided $t$ test. Data are presented as mean values ± S.D.. Scale bars, 1 cm.

3 overexpression delayed reentry into S phase after arrest at $G_0$/$G_1$ (Supplementary Fig. 4e). On the basis of these findings, we suggest that Gal-3 functions as a negative regulator of cell-cycle entry in HSCs and also actively maintains the balance of HSC quiescence and differentiation or division.

We further investigated changes in expression of the cell-cycle regulators cyclin-dependent kinase inhibitors (Cdkns). These include *p16* and *p21* family members that cause $G_0$/$G_1$ phase arrest and block entry into S phase[32]. We found that transcription of *p21* was reduced in purified Gal-3[−/−] LT-HSCs. In contrast, certain other Cdkns such as *p18* and *p27* which regulate the $G_1$/S transition, showed no significant differences (Fig. 3c). We also quantified p21 protein levels in Gal-3[+/+] or Gal-3[−/−] LT-HSCs by immunofluorescence staining. We observed that the expression of p21 protein was significantly lower in BM LT-HSCs of Gal-3[−/−] relative to Gal-3[+/+] mice (Fig. 3d, e). In addition, we analyzed the expression of *p16* and *p21* family members in BaF/Gal-3 cells. As expected, levels of *p21* and *p57* transcripts were higher in BaF/Gal-3 cells (Supplementary Fig. 4f). However, at the protein level, we confirmed the increase only of p21, but not p57, in these cells (Supplementary Fig. 4g). These results illustrate that deficiency of Gal-3 in LT-HSCs reduces p21 expression and prevents the maintenance of quiescence. In later analyses, we have focused on p21 because alteration of this protein is commonly observed in Gal-3-deficient LT-HSCs and Gal-3-overexpressing Ba/F3 cells.

Cheng et al. showed that p21 deficiency causes loss of quiescence in HSCs, resulting in hematopoietic disorders[17]. Hence, we employed the hematopoietic-stress model of 5-FU treatment, and investigated the relationship of Gal-3 deficiency and hematopoietic disorders in vivo. On treatment with a single dose of 5-FU, we observed a profound reduction of LT-HSC frequencies and numbers in BM from Gal-3[−/−] relative to Gal-3[+/+] mice (Fig. 3f, g). We further applied sequential 5-FU treatments to evaluate the survival rate in Gal-3[+/+] or Gal-3[−/−] mice. We found that Gal-3[−/−] mice died 2 weeks after the second injection, whereas although some Gal-3[+/+] mice died within 3 weeks, 70% survived until then (Fig. 3h). This suggests that hyper-proliferation of HSCs in Gal-3[−/−] mice enhances sensitivity to 5-FU and results in more toxicity to HSCs. These data are also consistent with the notion that Gal-3 deficiency in HSCs accelerates the cell cycle and reduces the number of quiescent cells.

Next, serial competitive BM-T was performed to assess how self-renewal and hematopoietic reconstitution capacities are affected by the lack of Gal-3. We intravenously transplanted 500 freshly-isolated LT-HSCs from Gal-3[+/+] or Gal-3[−/−] mice (CD45.2) into lethally-irradiated recipient mice (CD45.1) together with $5 \times 10^5$ competitor BM cells (CD45.1). After 16 weeks, we quantified the proportion of donor-derived LT-HSCs in BM and mature cells in peripheral blood from primary BM-T recipients. The results indicated no significant changes in self-renewal ability of HSCs and the ratio of B-cell, T-cell, and myeloid cells in peripheral blood was similar in both cases (Fig. 3i and Supplementary Fig. 4h). We then purified 500 donor-derived LT-HSCs from primary recipients and transplanted them into a second set of lethally-irradiated CD45.1[+] WT mice together with freshly-isolated CD45.1[+] competitors. Subsequent transplantations were performed in the same manner. We observed a significant reduction of donor-derived LT-HSCs and of their contribution to hematopoiesis in animals transplanted with Gal-3[−/−] LT-HSCs at the second and third sequential transfers, relative to mice receiving Gal-3[+/+] LT-HSCs (Fig. 3i and Supplementary Fig. 4h). This suggests that Gal-3[−/−] LT-HSCs were affected by hematopoietic-stress associated with repeated transplantation and that they gradually lost self-renewal capacity. Because of the great degree of LT-HSC phenotypic similarity in Gal-3[−/−] and p21[−/−] mice, it is suggested that Gal-3 may act as an upstream regulator of p21.

In summary, from these data we conclude that the key function of Gal-3 in LT-HSCs is to regulate cell-cycle arrest, thus maintaining quiescence of LT-HSCs, and that this function may be accomplished through regulating the expression of p21.

**Impaired cell-cycle progression of HSCs in Gal-3 Tg mice.** We showed above that proliferation and reentry into S phase after serum starvation arrest at $G_0$/$G_1$ was inhibited in BaF/Gal-3 cells (Supplementary Fig. 4d, e). To test whether LT-HSCs showed similar phenotypes in vivo, we generated transgenic (Tg) mice (Flox/Gal-3; Gal-3[flox/+]) expressing floxed *Gal-3* under the transcriptional control of the chicken β-actin (CAG) promoter. These were mated with mice expressing Cre recombinase under the direction of the *Tie2* or *Vav1* promoter (Tie2-Cre or Vav1-Cre) to generate Gal-3 Tg mice specifically in hematopoietic cells (Supplementary Fig. 5a). Survival rates suggested that both types of Gal-3 Tg mice (Tie2-Cre and Vav1-Cre) began to die from embryonic day (E) 10.5 onwards. No Gal-3 Tg embryos survived beyond E15.5 (Supplementary Fig. 5b). The gross appearance of E10.5 Gal-3 Tg mice indicated the presence of anemia in both embryo and yolk sac, and growth retardation was prevalent in both Tg mice (Supplementary Fig. 5c).

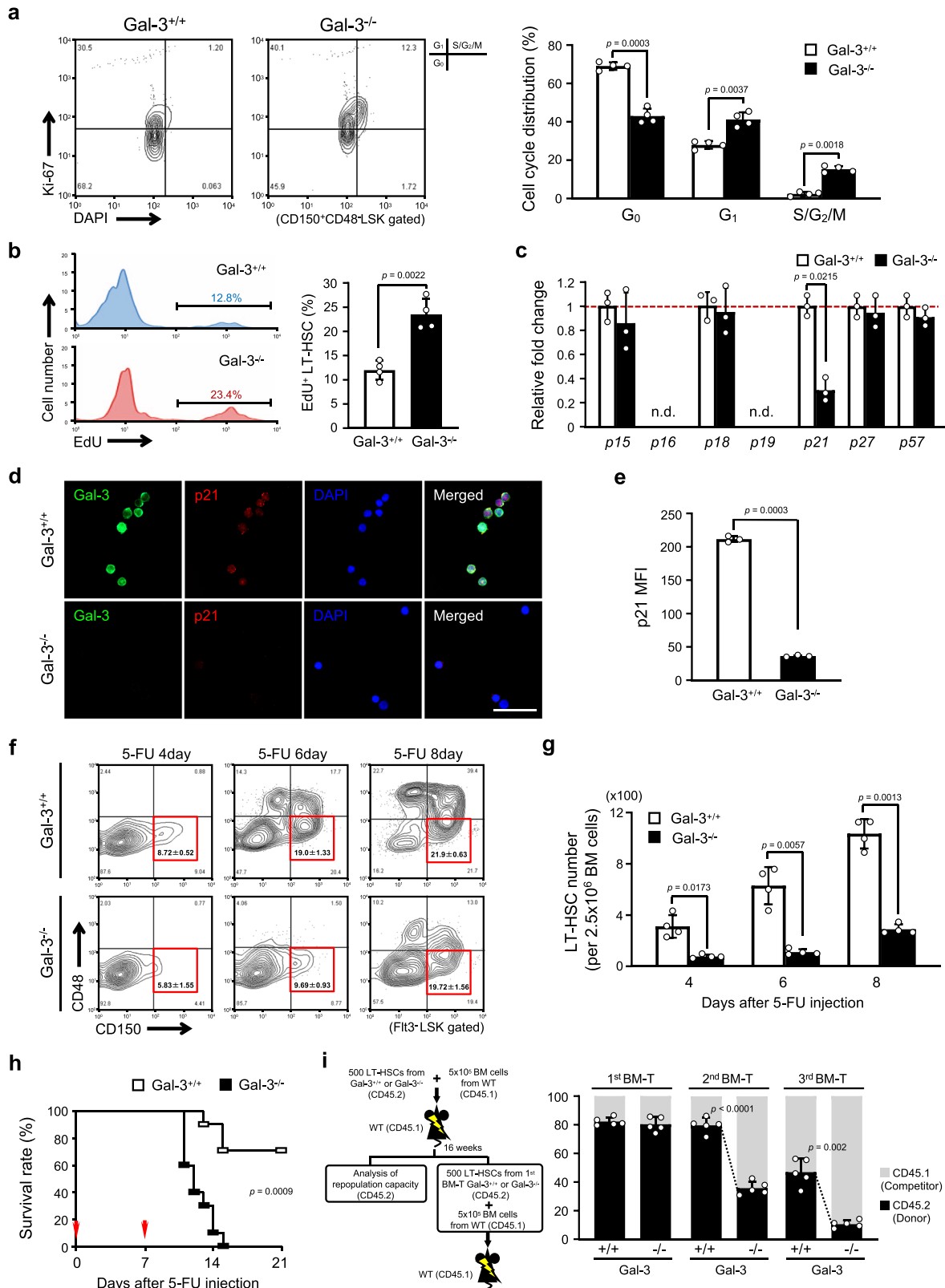

From around E10 onwards, Aorta gonad mesonephros (AGM)-derived HSCs colonize the fetal liver. The fetal site at which the first LT-HSC emerges is still under debate. Nevertheless, most investigators recognize that LT-HSC appear in the fetal liver at E11.5 where they greatly increase in number and initiate hematopoiesis[33,34]. Approximately at birth, HSCs start to migrate to the BM[35]. In order to highly purify LT-HSCs from E12.5 fetal liver, we opted to exploit the CD150-positive and CD48-negative phenotype of LSK cells[36]. Furthermore, for the purification of early stage (E10.5–E12.5) HSC/progenitors in fetal liver, we applied Lineage⁻c-Kit⁺AA4.1⁺ (LKAA4.1⁺) as a selection marker[37]. We isolated LT-HSCs from fetal liver of WT mice and Gal-3 Tg mice, and assessed Gal-3 mRNA expression. We found that Gal-3 expression was ≥300 times

**Fig. 3 Defect in the maintenance of LT-HSC quiescence in Gal-3$^{-/-}$ mice. a** (Left) Representative flow cytometric analysis showing the cell-cycle status of Gal-3$^{+/+}$ or Gal-3$^{-/-}$ LT-HSCs (CD150$^+$CD48$^-$LSK) using Ki-67 and DAPI staining. (Right) Bar graph showing the percentage of cells in G$_0$, G$_1$, and S/G$_2$/M phases of the cell cycle ($n = 4$ biological replicates per genotype, compared by two-sided $t$ test). Data are presented as mean values ± S.D. **b** (Left) Representative flow cytometry histograms for frequencies of EdU-positive LT-HSC (CD150$^+$CD48$^-$Flt3$^-$LSK) in the BM of Gal-3$^{+/+}$ or Gal-3$^{-/-}$ mice. (Right) Bar graph showing the percentage of EdU-positive cells ($n = 4$ biological replicates per genotype, compared by two-sided $t$ test). Data are presented as mean values ± S.D. **c** Relative expression level of mRNA for *p16* and *p21* family members in BM LT-HSCs (CD150$^+$CD48$^-$Flt3$^-$LSK) from Gal-3$^{+/+}$ or Gal-3$^{-/-}$ mice ($n = 3$ biological replicates per genotype, compared by two-sided $t$ test). Data are presented as mean values ± S.D. and n.d. not determined. **d** Representative immunofluorescence images of LT-HSCs (CD150$^+$CD48$^-$Flt3$^-$LSK) labeled by Gal-3 (green) and p21 (red) in BM of Gal-3$^{+/+}$ or Gal-3$^{-/-}$ mice ($n = 3$ biological replicates per genotype). DAPI (blue) was used to detect nuclei. Scale bar, 50 μm. **e** Bar graph showing the mean fluorescence intensity (MFI) of p21 in LT-HSCs (CD150$^+$CD48$^-$Flt3$^-$LSK) in BM of Gal-3$^{+/+}$ or Gal-3$^{-/-}$ mice. The MFI was quantified in seven random fields of three independent cell specimens per genotype using Image J software (two-sided $t$ test). Data are presented as mean values ± S.D. **f** Kinetics of LT-HSCs (CD150$^+$CD48$^-$Flt3$^-$LSK) changes after 5-FU treatment. Representative data from flow cytometric analysis and quantitative evaluation of frequency are shown ($n = 4$ biological replicates per genotype at each time point, compared by two-sided $t$ test). Data are presented as mean values ± S.D. and exact $p$ values: 5-FU 4 day: $p = 0.0196$ vs. Gal-3$^{-/-}$; 5-FU 6 day: $p = 0.0013$ vs. Gal-3$^{-/-}$; 5-FU 8 day: $p = 0.1034$ vs. Gal-3$^{-/-}$. **g** Absolute number of LT-HSCs (CD150$^+$CD48$^-$Flt3$^-$LSK) in BM of Gal-3$^{+/+}$ or Gal-3$^{-/-}$ mice at different time points after 5-FU treatment ($n = 4$ biological replicates per genotype at each time point, compared by two-sided $t$ test). Data are presented as mean values ± S.D. **h** Survival rate of Gal-3$^{+/+}$ or Gal-3$^{-/-}$ mice following sequential 5-FU treatment. Red arrows indicate 5-FU injection ($n = 10$ biological replicates per genotype, compared by two-sided $t$ test). Data are presented as mean values ± S.D. **i** Defective long-term reconstitution capacity of Gal-3$^{-/-}$ LT-HSCs during serial competitive BM-T. (Left) Experimental schema for serial competitive BM-T. (Right) The percentage of Gal-3$^{+/+}$ or Gal-3$^{-/-}$ donor-derived LT-HSCs (CD45.1$^-$CD150$^+$CD48$^-$Flt3$^-$LSK) was determined at 16 weeks post-transplant for the primary, secondary and third BM-Ts ($n = 5$ biological replicates per genotype at each BM-T time point, compared by two-sided $t$ test). Data are presented as mean values ± S.D.

higher in fetal liver LT-HSCs of Gal-3 Tg mice than WT or control mice (Flox/Gal-3) (Fig. 4a).

Next, we investigated how Gal-3 overexpression affects the development of HSCs. The number of fetal liver HSC/progenitors (LKAA4.1$^+$) in embryos gradually increased from E10.5 to E12.5 in controls, but was reduced in Gal-3 Tg embryos on both the Tie2-Cre and Vav1-Cre background (Fig. 4b). Because Tie2 is expressed in ECs as well as HSCs, it is possible that Gal-3 overexpression in ECs affects HSC development indirectly. Therefore, we examined whether growth retardation of HSCs is cell autonomous. We found no significant differences between vascular formation in control and Gal-3 Tg mice (Tie2-Cre background) at E9.5 (Supplementary Fig. 5d). However, fetal liver HSCs (LSK) overexpressing Gal-3 had lower in vitro hematopoietic colony-forming capacity than controls (Fig. 4c). In addition, we found that Gal-3 overexpression did not affect specific differentiation into different lineages (Fig. 4d), suggesting that disordered hematopoiesis was due to the suppression of cell division in HSCs. We also analyzed apoptosis and cell-cycle entry in fetal liver LT-HSCs and found that percentages of apoptotic cells amongst LT-HSCs from Gal-3 Tg embryos (as estimated by Annexin V and SYTOX Green staining), were not different from control embryos (Supplementary Fig. 5e). However, cell-cycle analyses (by Ki-67 and DAPI staining) suggested that a large proportion of fetal liver LT-HSCs from Gal-3 Tg embryos remained in a quiescent G$_0$ state, with both G$_1$ and S/G$_2$/M phases decreased (Fig. 4e). To determine cell-cycle status precisely, we isolated fetal liver LT-HSCs and measured DNA synthesis by labeling with EdU. The proportion of actively dividing HSCs among all LT-HSCs was significantly lower in Gal-3 Tg than in control mice, suggesting that proliferation was impaired in the former (Supplementary Fig. 5f). Furthermore, we also assessed *p21* and *p16* family genes in these fetal liver LT-HSCs and found that *p21* was more strongly expressed in the Gal-3 Tg mice than in controls (Fig. 4f).

To evaluate the effect of Gal-3 overexpression in adult LT-HSCs, we performed recombination experiments between two LoxP sites in LT-HSCs by Cre Recombinase Gesicles ex vivo (Fig. 4g). We co-cultured purified LT-HSCs from the BM of Flox-CAT-EGFP or Flox/Gal-3 mice with Cre Recombinase Gesicles for 4 h. Subsequently, recombination efficiency was evaluated after 20 h of culture. This approach revealed that >70% of GFP$^+$

cells were present in LT-HSCs of Flox-CAT-EGFP mice and there were >90% mCherry$^+$ cells in LT-HSCs of Flox/Gal-3 mice after Cre Recombinase Gesicle treatment. This suggests that ex vivo Cre Recombinase Gesicle treatment achieved a high recombination efficiency (Fig. 4g, Supplementary Fig. 5g). We used this method to treat WT or Flox/Gal-3 LT-HSCs and assessed the level of expression of Gal-3. As expected, *Gal-3* expression in Flox/Gal-3-derived LT-HSCs was >30-fold higher than in WT mouse-derived LT-HSCs (Fig. 4h). We found that overexpression of Gal-3 resulted in significantly decreased colony-forming ability of adult LSK cells (Fig. 4i), but did not affect the proportions of each cell type in the colonies (Fig. 4j). We next investigated the cell-cycle status of LT-HSCs after Cre Recombinase Gesicle treatment, and found that Gal-3 overexpression led to an increase in the percentage of cells in the G$_0$ phase (Fig. 4k). Similarly, Gal-3 overexpression led to a decreased rate of EdU incorporation into LT-HSCs in vitro, suggesting that Gal-3 inhibits DNA synthesis in these cells (Fig. 4l). Further, we analyzed the expression of members of the *p16* and *p21* family in LT-HSCs from the BM of WT or Flox/Gal-3 mice after Cre Recombinase Gesicle treatment. These experiments showed that Gal-3 over-expression enhanced the expression of *p21* (Fig. 4m).

Taken together, these results indicate that overexpression of Gal-3 in LT-HSC can induce cell-cycle arrest, leading to inhibition of proliferation. In early stages of embryonic development, dysregulation of the cell cycle in HSCs can cause anemia and lethality.

**Ang-1/Tie2 or Thpo/Mpl signaling regulates *Gal-3* expression.** Next, we investigated the regulatory mechanisms controlling Gal-3 expression in LT-HSCs. We examined the effects of Ang-1, Thpo, and CXCL12, which are well known to induce HSC dormancy[19],[20],[38]. We found that on Ang-1 or Thpo stimulation, *Gal-3* mRNA expression gradually increased in a time-dependent manner (Fig. 5a, b), but there was no response to CXCL12 (Fig. 5c). These results suggest that transcriptional up-regulation of *Gal-3* is mediated by Ang-1/Tie2 or Thpo/Mpl signaling. Hence, our research has focused on the common downstream elements of Ang-1/Tie2 and Thpo/Mpl signaling pathways. Previous work had indicated that Tie2 or Mpl activated the downstream signaling effector molecules PI3K and MEK[39]. We investigated whether inhibition of PI3K or MEK signaling

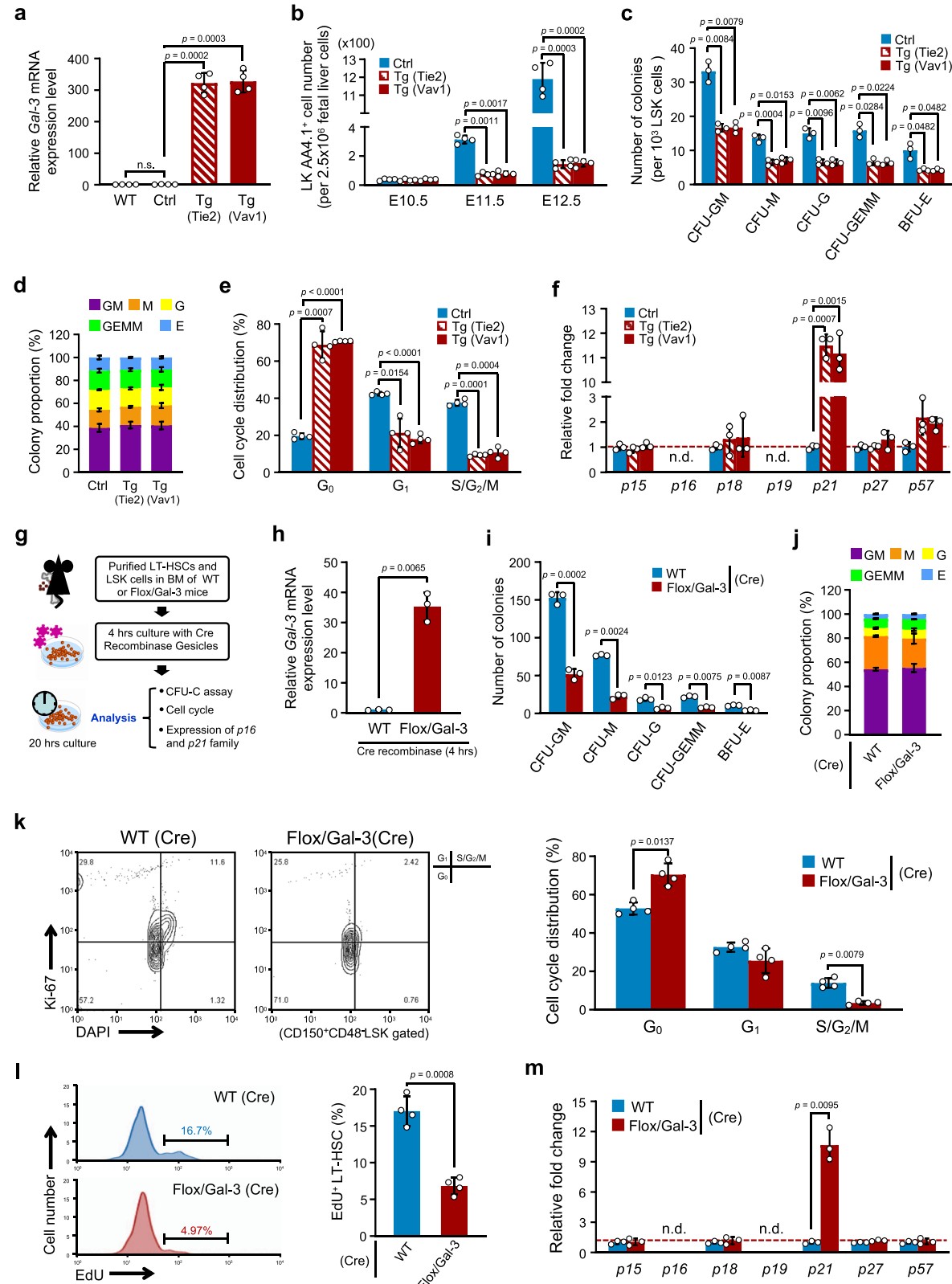

pathways affects the expression of Gal-3 in LT-HSCs. Sorted LT-HSCs from BM were pre-treated short-term with specific inhibitors of PI3K or MEK, and then stimulated with Ang-1 or Thpo (Fig. 5d). We found that the PI3K inhibitor LY294002 but not the MEK inhibitor PD98059 blocked Ang-1- or Thpo-induced Gal-3 expression (Fig. 5e, f), suggesting a role for the PI3K signaling pathway in the transcriptional regulation of Gal-3. Several lines of

evidence had previously implicated involvement of the transcription factor NF-κB in Gal-3 regulation[24,40], and suggested that NF-κB translocation is activated via the PI3K-AKT pathway[41]. Therefore, we also tested whether AKT was involved in our system and found that AKT phosphorylation increased in LT-HSCs relative to controls on Ang-1 or Thpo stimulation (Fig. 5g). Hence, we examined the effect of Ang-1 or Thpo

**Fig. 4 Impaired cell-cycle progression of Gal-3-overexpressing HSC. a** Relative expression level of *Gal-3* mRNA in fetal liver LT-HSCs (CD150$^+$CD48$^-$LSK) from E12.5 WT, control (Ctrl, Flox/Gal-3) and Gal-3 Tg (Tie2-cre-Flox/Gal-3 or Vav-1-Cre-Flox/Gal-3) embryos ($n = 4$ biological replicates per genotype, compared by two-sided *t* test). Data are presented as mean values ± S.D. and n.s. not significant. **b** Absolute number of fetal liver HSC/progenitors (Lin$^-$ c-Kit$^+$ AA4.1$^+$) from Ctrl or Gal-3 Tg embryos at different gestational ages ($n = 4$ biological replicates per genotype, compared by two-sided *t* test). Data are presented as mean values ± S.D. **c** Colony-forming potential of fetal liver HSCs (LSK) from E12.5 Ctrl and Gal-3 Tg embryos in CFU-C assays ($n = 3$ biological replicates per genotype, compared by two-sided *t* test). Data are presented as mean values ± S.D. **d** Percentages of different colony types developed as described in **c** ($n = 3$ biological replicates per genotype). **e** Cell-cycle analysis of Ctrl and Gal-3 Tg fetal liver LT-HSCs (CD150$^+$CD48$^-$LSK) from E12.5 embryos by staining with Ki-67 and DAPI. The percentage of cells in G$_0$, G$_1$, and S/G$_2$/M phases is shown in the bar graph ($n = 4$ biological replicates per genotype, compared by two-sided *t* test). Data are presented as mean values ± S.D. **f** Relative expression level of mRNA for *p16* and *p21* family members in fetal liver LT-HSCs (CD150$^+$CD48$^-$LSK) from E12.5 Ctrl and Gal-3 Tg embryos ($n = 3$ biological replicates per genotype, compared by two-sided *t* test). Data are presented as mean values ± S.D. and n.d. not determined. **g** Experimental schema for Cre Recombinase Gesicles-inducible ex vivo Gal-3 overexpression. **h** Relative expression level of *Gal-3* mRNA in LT-HSCs (CD150$^+$CD48$^-$Flt3$^-$LSK) in BM of WT or Flox/Gal-3 mice following Cre Recombinase Gesicle treatment ($n = 3$ biological replicates per genotype, compared by two-sided *t* test). Data are presented as mean values ± S.D. **i** Colony-forming potential of LSK cells in BM of WT or Flox/Gal-3 mice following Cre Recombinase Gesicle treatment in CFU-C assays ($n = 3$ biological replicates per genotype, compared by two-sided *t* test). Data are presented as mean values ± S.D. **j** Percentages of different colony types developed as described in (**i**) ($n = 3$ biological replicates per genotype). **k** (Left) Representative flow cytometric analysis showing the cell-cycle status of WT or Flox/Gal-3 LT-HSCs (CD150$^+$CD48$^-$LSK) following Cre Recombinase Gesicle treatment using Ki-67 and DAPI staining. (Right) Bar graph showing the percentage of cells in G$_0$, G$_1$, and S/G$_2$/M phases of cell-cycle ($n = 4$ biological replicates per genotype, compared by two-sided *t* test). Data are presented as mean values ± S.D. **l** (Left) Representative flow cytometry histograms for EdU-positive LT-HSC (CD150$^+$CD48$^-$Flt3$^-$LSK) frequencies in BM of WT or Flox/Gal-3 mice following Cre Recombinase Gesicle treatment. (Right) Bar graph showing the percentage of EdU-positive cells ($n = 4$ biological replicates per genotype, compared by two-sided *t* test). Data are presented as mean values ± S.D. **m** Relative expression level of mRNA for *p16* and *p21* family members in BM LT-HSCs (CD150$^+$CD48$^-$Flt3$^-$LSK) from WT or Flox/Gal-3 mice following Cre Recombinase Gesicle treatment ($n = 3$ biological replicates per genotype, compared by two-sided *t* test). Data are presented as mean values ± S.D. and n.d. not determined.

stimulation on nuclear translocation of NF-κB (p65) in LT-HSCs. As expected, we found that Tie2 or Mpl activation by Ang-1 or Thpo promoted translocation of cytoplasmic NF-κB into the nucleus within 4 h (Fig. 5h, i). To determine whether inhibiting NF-κB activity would reduce *Gal-3* expression in LT-HSCs, we isolated LT-HSCs from WT BM and stimulated them with Ang-1 or Thpo in the presence or absence of the NF-κB antagonist BAY-11-7082 to investigate effects on *Gal-3* expression after 8 h (Fig. 5j). We found that *Gal-3* mRNA induction by Ang-1 or Thpo was reduced by NF-κB inhibition in a dose dependent manner (Fig. 5k, l). Furthermore, we analyzed the frequency of LT-HSCs, and *Gal-3*, *p21* expression in LT-HSC 12 h after in vivo treatment with the NF-κB antagonist BAY-11-7082 (Fig. 5m). We found that this resulted in a decreased the proportion of LT-HSCs and was accompanied by down-regulation of the mRNA levels of *Gal-3* and *p21* in these cells (Fig. 5n, o).

Taken together, from these data we conclude that in the BM microenvironment, Ang-1 or Thpo stimulates Tie2 or Mpl respectively, and activates the PI3K-AKT signaling pathway. Subsequently, *Gal-3* expression is induced following translocation of NF-κB to the nucleus.

**Gal-3 regulates *p21* transcription in LT-HSCs.** Finally, we assessed how Gal-3 regulates the expression of *p21* in LT-HSCs. Accumulating evidence currently suggests that Gal-3 induces p21 expression in tumor cell lines, such as BT549 (human breast cancer) and DU145 (human prostate cancer), associated with cell-cycle arrest[42,43]. However, the exact underlying mechanism is still unknown. Therefore, we examined how Gal-3 regulates *p21* transcription in LT-HSCs. The transcription of this gene can be activated through both p53-dependent and p53-independent mechanisms[44]. Thus, we examined the expression of *p53* in LT-HSCs from Gal-3$^{+/+}$ or Gal-3$^{-/-}$ mice, but found no differences at the transcription level in purified Gal-3$^{+/+}$ and Gal-3$^{-/-}$ LT-HSCs (Supplementary Fig. 6a). In addition, there were no differences in *p53* mRNA and protein expression between Ba/F3 cells expressing Gal-3 and BaF/mock cells (Supplementary Fig. 6b, c). To further explore interactions between Gal-3 and p53 in LT-HSCs, we assessed hematopoietic phenotypes in p53-deficient mice (p53$^{-/-}$). First, we quantified the percentages and numbers of LT-HSCs, ST-HSCs, and the MPPs2-4 population in

p53$^{-/-}$ mice. Neither percentages nor numbers of these cells were affected by the absence of p53 (Supplementary Fig. 6d, e). There were also no differences between p53$^{+/+}$ and p53$^{-/-}$ HSCs in the number of CFU-C generated in vitro (Supplementary Fig. 6f). Next, we investigated whether cell-cycle effectors of the *p16* and *p21* families were affected by the lack of p53 expression. The levels of these molecules were found to be normal in p53$^{-/-}$ LT-HSCs (Supplementary Fig. 6g). Taken together, we concluded from these data that Gal-3 does not interact with p53 and that *p21* transcription is regulated by Gal-3 independent of p53.

Hence, we used ChIP assays to seek possible binding regions for Gal-3 on *p21* promoters. We found that Gal-3 possibly binds at the −1 to −302 region of the *p21* promoter (Fig. 6a). This is where the binding sites of transcription factor Sp1 are located, suggesting that Gal-3 might form a complex with Sp1. To investigate this possibility, we used Ba/F3 cells which express Sp1 regardless of Gal-3 overexpression (Fig. 6b). Co-immunoprecipitation analysis showed that Sp1 did interact with Gal-3 in BaF/Gal-3 cells (Fig. 6c). We found that knock-down of Sp1 by siRNA resulted in a significant decrease of p21 in these cells (Fig. 6d). Next, to validate the association of Sp1 with p21 in LT-HSC, we analyzed the expression and location of Sp1 in LT-HSCs using immunofluorescence. We observed that Sp1 was highly expressed in the nucleus of LT-HSC and co-localized with Gal-3 (Fig. 6e), suggesting their physical interaction in these cells. In addition, in order to assess whether Sp1 inhibition decreased the expression of *p21* also in vivo, we administered the Sp1 inhibitors Mithramycin A (MIT) and Tolfenamic Acid (TA) together with Verapamil to block the ABC transporters abundantly expressed by LT-HSCs (Fig. 6f). Treatment with MIT and TA decreased frequency of LT-HSC within the Flt3$^-$LSK population, similar to that seen in Gal-3$^{-/-}$ mice (Fig. 6g). We also found that *p21* mRNA was decreased on treatment with MIT and TA (Fig. 6h). This indicates that LT-HSCs had escaped the quiescent state and that cell-cycle entry was accelerated. These data suggest that Sp1 acts as an important factor in p21 regulation, together with Gal-3.

## Discussion

In this study, we clarified the main functions and regulation of Gal-3 in HSCs and identified a molecular mechanism important

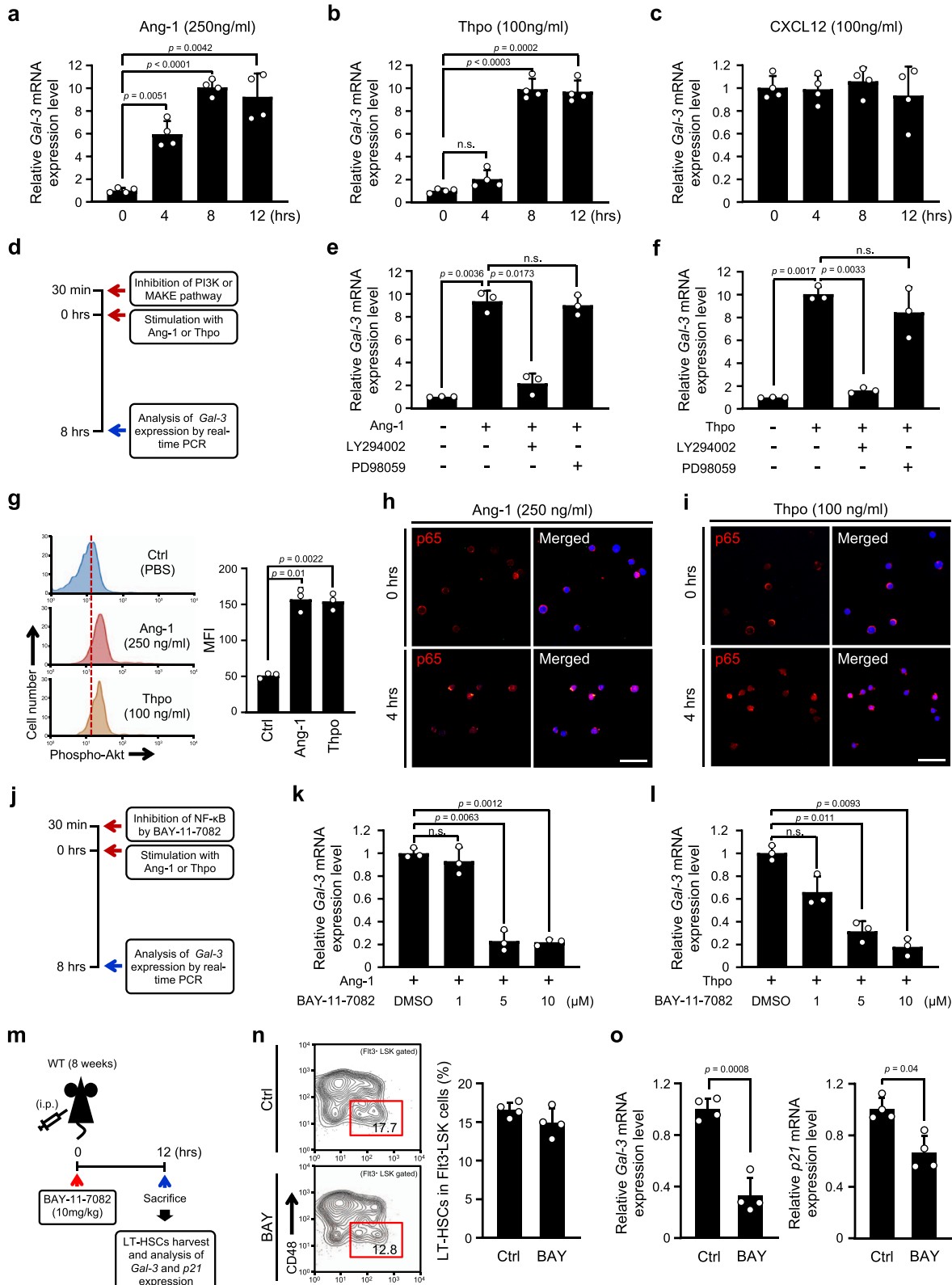

for the maintenance of stemness in HSCs. We found that nuclear translocation of NF-κB mediated by activated AKT following stimulation by Tie2 or Mpl expressed in HSCs promotes the production of Gal-3, which then binds to Sp1 and induces *p21* transcription, which results in the inhibition of cell-cycle progression and maintains HSC quiescence (Fig. 6i).

**Extrinsic molecular cues for Gal-3 expression**. It is of interest to understand the interacting signals, i.e., ligand-receptor pairs in the niche that maintain HSCs in a quiescent state. It is believed that interactions between HSCs and niche cells is crucial for the long-term maintenance of HSC quiescence. We previously reported that Tie2 is expressed on HSCs when they first emerge in

**Fig. 5 Transcriptional regulation of *Gal-3* expression mediated by Ang-1/Tie2 or Thpo/Mpl signaling through the PI3K/AKT-NFkB pathway. a–c**
Relative expression level *Gal-3* mRNA in LT-HSCs (CD150$^+$CD48$^-$Flt3$^-$LSK) in BM of WT mice after stimulation with Ang-1 (**a**), Thpo (**b**), and CXCL12 (**c**) ($n = 4$ biological replicates, compared by two-sided *t* test). Data are presented as mean values ± S.D. and n.s. not significant. **d** Experimental schema for ex vivo inhibition of the PI3K/AKT or MEK/ERK pathway. **e, f** Analysis of Tie2 and Mpl downstream signaling pathways for the expression of Gal-3. WT LT-HSCs (CD150$^+$CD48$^-$Flt3$^-$LSK) were preincubated with either PD98059 (50 μM, MEK inhibitor), LY294002 (10 μM, PI3K inhibitor), or DMSO (control, 0.05%) for 30 min before Ang-1 (250 ng/ml, **e**) or Thpo (100 ng/ml, **f**) stimulation for 8 h. The samples were analyzed by real-time PCR ($n = 3$ biological replicates, compared by two-sided *t* test). Data are presented as mean values ± S.D. and n.s. not significant. **g** (Left) Representative flow cytometry histogram showing changes in AKT phosphorylation between Ang-1- or Thpo-treated and PBS-treated LT-HSCs (CD150$^+$CD48$^-$Flt3$^-$LSK). (Right) Quantification of AKT phosphorylation expressed as arbitrary units of the mean fluorescence intensity (MFI a.u.) in LT-HSCs after stimulation with Ang-1 or Thpo ($n = 3$ biological replicates, compared by two-sided *t* test). Data are presented as mean values ± S.D. **h, i** Nuclear translocation of p65 (NF-κB) in response to Tie2 or Mpl activation. LT-HSCs (CD150$^+$CD48$^-$Flt3$^-$LSK) were treated with Ang-1 (**h**) or Thpo (**i**) and stained with anti-p65 (red) and DAPI (blue) antibodies. A representative image is shown ($n = 3$ biological replicates). Scale bar, 50 μm. **j** Experimental schema for ex vivo BAY-11-7082 (NF-κB inhibitor) treatment. **k, l** Relative expression level of *Gal-3* mRNA in WT LT-HSCs (CD150$^+$CD48$^-$Flt3$^-$LSK) on ex vivo BAY-11-7082 treatment. LT-HSCs were preincubated with BAY-11-7082 or DMSO (0.05%) for 30 min before Ang-1 (250 ng/ml, **k**) or Thpo (100 ng/ml, **l**) stimulation for 8 h ($n = 3$ biological replicates, compared by two-sided *t* test). Data are presented as mean values ± S.D. and n.s. not significant. **m** Experimental schema for in vivo BAY-11-7082 treatment. **n** (Left) Representative flow cytometric analysis of LT-HSC (CD150$^+$CD48$^-$Flt3$^-$LSK) frequencies (red box) in BM of PBS-treated (Ctrl) or BAY-11-7082-treated (BAY) mice. (Right) Bar graph showing the percentage of LT-HSCs in the Flt3$^-$LSK population ($n = 4$ biological replicates). **o** Relative expression level *Gal-3* and *p21* mRNA in LT-HSCs (CD150$^+$CD48$^-$Flt3$^-$LSK) in BM of PBS-treated (Ctrl) or BAY-11-7082-treated (BAY) mice. Cells were harvested at 12 h ($n = 4$ biological replicates, compared by two-sided *t* test). Data are presented as mean values ± S.D.

the mouse embryo and that it plays an essential role for their colonization in the omphalomesenteric artery[45]. Furthermore, adhesiveness of Tie2$^+$ LSK cells to extracellular matrix and cellular components such as osteoblasts is induced by Ang-1/Tie2 signaling mediated by activation or expression of β1-integrin and N-cadherin[19]. However, recent research indicated that N-cadherin-deficiency in osteoblasts did not affect HSC dormancy[46,47]. Moreover, using Ang-1-deficient mice (Col1a1*2.3-Cre; Angpt1$^{fl/fl}$, Lepr-Cre; Angpt1$^{fl/fl}$, and Mx1-Cre; Angpt1$^{fl/fl}$) it was reported that Ang-1 in the BM microenvironment is not involved in the regulation of HSC quiescence[48]. These data suggest that Ang-1/Tie2-mediated HSC quiescence is controlled in other ways. Our results reported here suggest that lack of Ang-1/Tie2 may be compensated by Thpo/Mpl in HSCs by mechanisms involving Gal-3, consistent with findings that Thpo- or Mpl-deficient mice have fewer HSCs in the BM[49,50]. Thpo is secreted by niche cell components such as osteoblasts and megakaryocytes. Although we could not distinguish in vivo dominance of Gal-3 expression in HSCs caused by Ang-1 and Thpo, further data are required to determine which extrinsic molecular cues other than Ang-1 and Thpo from niche cells are critical for Gal-3 expression.

**Heterogeneous Gal-3 expression in LT-HSCs.** Using immunofluorescence staining, we found heterogeneous Gal-3 expression in different cell types (Fig. 1b). Recently, LT-HSC-related research revealed that there is a heterogeneity in LT-HSCs with at least five clusters proposed. Cell in these different clusters exhibit clearly different metabolic status[51]. We conjecture that altered Gal-3 expression levels regulate the activation and quiescence of LT-HSC in these five different individual clusters. We applied Image J to quantify Gal-3 protein expression in the nuclear area of these different cell types (LT-HSC, ST-HSC, and MPPs2-4). The results were consistent with the notion that the function of Gal-3 is mainly concerned with nuclear activity of LT-HSCs and are also consisted with our conclusion that Gal-3 regulates the transcription of *p21* and promotes the quiescence of LT-HSCs.

**Gal-3 and inhibitors of cyclin-dependent kinases.** It is widely believed that p21 and p57 are negative regulators of the cell cycle in HSCs[17,18]. In the present study, we found that overexpression of Gal-3 in HSCs induced p21 and caused cell-cycle arrest. Accumulation of p21 is controlled by the tumor suppressor p53

and is well known to induce cell-cycle arrest in many different cell types[52]. However, based on the analysis of Gal-3 function in p53-deficient HSCs, we found no relationship between Gal-3 and p53.

Previous studies using cancer cells and hepatocytes suggested that Gal-3 induces p21 expression, resulting in negative regulation of the cell cycle[43,44,53]. In our analysis, although *p21* was significantly upregulated in Ba/F3 cells by Gal-3, *p57* expression was only slightly increased (Supplementary Fig. 4f). However, as far as we could determine by immunofluorescence, p57 protein levels in LT-HSCs from Gal-3$^{-/-}$ mice were not significantly different from WT mice. It has been suggested that decreased *p57* transcription is caused by the lack of Gal-3-Sp1 complex binding, because Sp1 binding sites are also present in the *p57* promoter region[54,55]. However, our results showed that Gal-3 is not involved in p57 protein expression in any marked manner (Supplementary Fig. 4g).

**Gal-3 and inflammatory responses of HSC.** According to a previous report, Gal-3$^{-/-}$ mice have weaker responses to inflammatory stimuli in the peritoneal cavity[28]. Hence, we analyzed the effect of lipopolysaccharide (LPS)-induced inflammation on LT-HSCs in the BM of Gal-3$^{+/+}$ or Gal-3$^{-/-}$ mice. We found that neither the fold-change of percentages or absolute numbers of BM LT-HSC showed any significant differences between Gal-3$^{+/+}$ and Gal-3$^{-/-}$ mice (Supplementary Fig. 7a, b). LPS treatment indeed induced the proliferation of LT-HSC which returned to baseline after 72 h, but there were also no differences between Gal-3$^{+/+}$ and Gal-3$^{-/-}$ mice in this respect. Because LPS-induced inflammation is an acute inflammatory response, its effect on the hematopoietic system is transient. Chronic inflammation or aging may have more profound effects on the hematopoietic system, especially on HSCs lacking Gal-3. We need further data on how Gal-3 deficiency affects LT-HSC in such situations.

**Requirement for monitoring Gal-3 protein expression in further analyses.** We found that intracellular Gal-3 is crucial for cell-cycle regulation in HSCs. Isolation of Gal-3$^+$ HSC populations using other stem-cell markers in flow cytometry is not currently an option. Reporter mice harboring fluorescent protein under the transcriptional control of the *Gal-3* promoter are not suitable because transcription levels may not correctly reflect the status of HSCs due to lack of transcriptional activation in dormant HSCs.

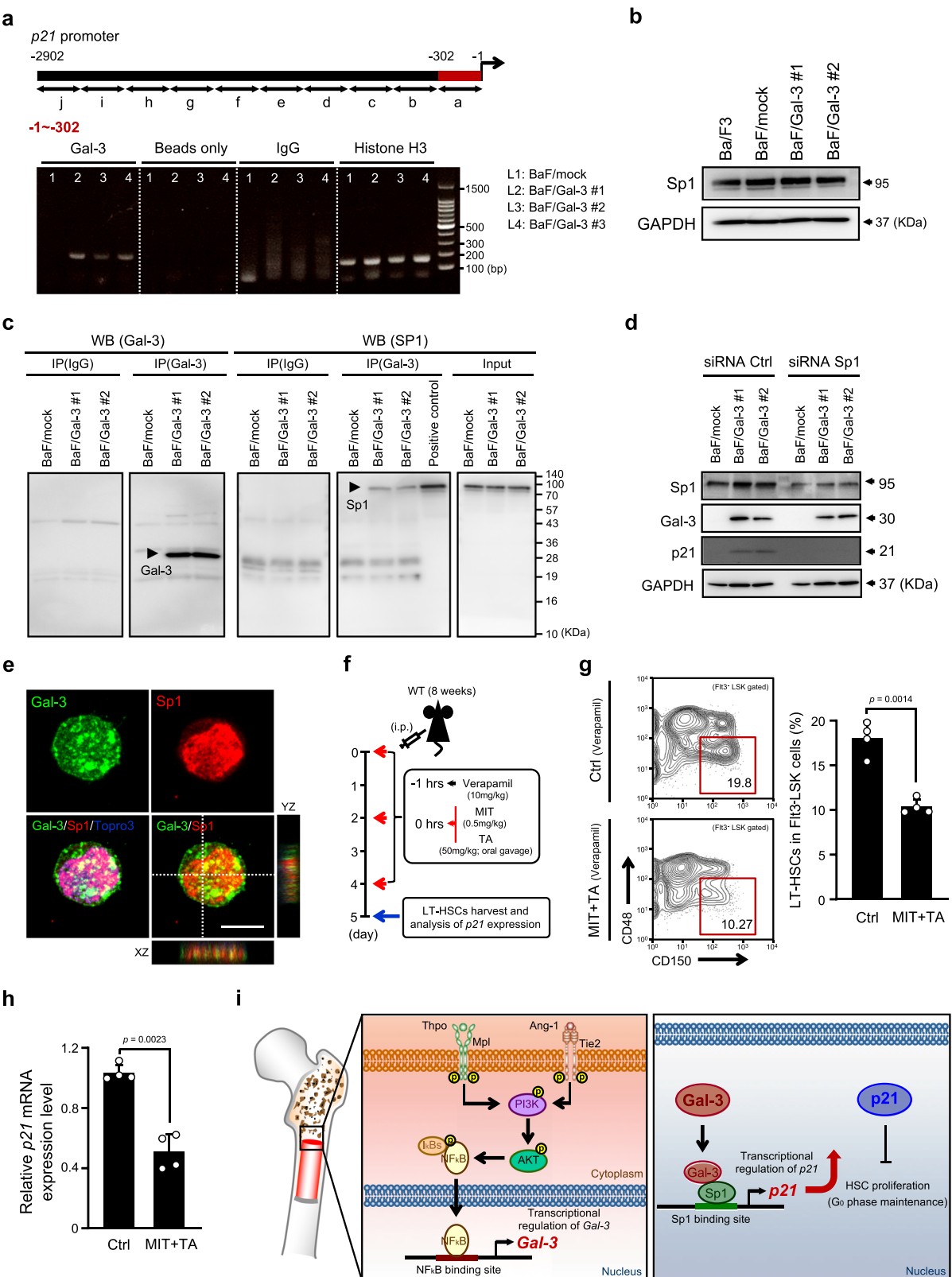

Therefore, Gal-3 molecular tagging is necessary for isolating Gal-3+ HSCs in order to analyze the cell-cycle precisely. To analyze the stemness of HSCs in the presence or absence of Gal-3, precise in vitro analysis using Gal-3-positive or -negative HSCs is likely to be critical. Precise functional analysis using Gal-3 molecularly tagged mice will be valuable in the future.

## Methods

**Mice**. C57BL/6 mice (Ly5.2; Gal-3+/+) were purchased from Japan SLC (Shizuoka, Japan). C57BL/6-Ly5.1 mice were purchased from Sankyo Labo Service (Tsukuba, Japan). Gal-3-knockout (Ly5.2; Gal-3−/−) mice[28], Tie2-Cre mice[56] and Flox-CAT-EGFP mice[57] have been reported elsewhere. A2Kio/J (Vav1-iCre) and Trp53tm1Tyj/J (p53+/−) mice were purchased from The Jackson Laboratory (Bar Harbor, ME). Eight-12 weeks old mice (female) were used for all experiments. Animals were

**Fig. 6 Gal-3 regulates *p21* transcriptional activation in LT-HSCs. a** ChIP analysis of Gal-3 binding to the *p21* promoter in BaF/Gal-3 cells. (Top) Schematic diagram representing the binding region of the *p21* promoter (−1 to −302). (Bottom) Quantification of signal intensities of the PCR products. A representative image is shown (*n* = 3 biologically independent experiments). Anti-histone H3, normal anti-rabbit IgG, and beads were also used as positive controls, negative controls, and blanks in each experiment. **b** Western blotting of Sp1 in BaF3, BaF/mock, and BaF/Gal-3 cells (#1, #2). **c** Protein complexes isolated by immunoprecipitation from BaF/mock and BaF/Gal-3 cells (#1, #2) using anti-Gal-3 and IgG (control) antibodies. Immunoprecipitates were analyzed by western blotting with anti-Gal-3 and anti-Sp1 antibodies. **d** Western blotting of Gal-3 and p21 in Sp1-silenced BaF/mock or Sp1-silenced BaF/Gal-3 cells (#1, #2). Representative Sp1 protein silencing blots of four different Sp1-specific-siRNAs and control siRNA. **e** Co-localization of Gal-3 (green) and Sp1 (red) proteins in nuclei of freshly-isolated LT-HSCs (CD150$^+$CD48$^-$Flt3$^-$LSK) from the BM of WT mice. TO-PRO-3 (blue) was used to stain nuclei showing Z stacks in cells as depicted by the dashed line. A representative image is shown (*n* = 4 biological replicates). Scale bar, 10 μm. **f** Experimental schema for in vivo Mithramycin A (MIT) and Tolfenamic Acid (TA) treatment. Verapamil was used to increase the efficiency of the Sp1 inhibitor in LT-HSCs. **g** (Left) Representative flow cytometric analysis of LT-HSC (CD150$^+$CD48$^-$Flt3$^-$LSK) frequencies (red box) in BM of PBS-treated (Ctrl) or MIT + TA treated mice. (Right) Bar graph showing the percentage of LT-HSCs in the Flt3$^-$LSK population (*n* = 4 biological replicates, compared by two-sided *t* test). Data are presented as mean values ± S.D. **h** Relative expression level of *p21* mRNA in LT-HSCs (CD150$^+$CD48$^-$Flt3$^-$LSK) in BM of PBS-treated (Ctrl) or MIT + TA treated mice (*n* = 4 biological replicates, compared by two-sided *t* test). Data are presented as mean values ± S.D. **i** Proposed model showing the molecular mechanisms essential for maintaining LT-HSC quiescence. Nuclear translocation of NF-κB mediated by activated AKT following stimulation by Tie2 or Mpl expressed in LT-HSCs promotes the production of Gal-3, which then binds to Sp1 and induces *p21* transcription, which in turn results in the inhibition of cell-cycle progression and maintenance of LT-HSC quiescence.

housed in environmentally controlled rooms of the animal experimentation facility at Osaka University. All experiments were performed in accordance with the guidelines of Osaka University Committee for Animal and Recombinant DNA Experiments (Approval number: 4062). Mice were handled and maintained according to Osaka University guidelines for animal experimentation.

**Generation of Flox/Gal-3 Mice.** Full-length Gal-3 cDNA was isolated from intestinal total RNA using PCR-based cloning methods[58]. Primers used for cloning were as follows: sense: 5′-GGA ATT CGG ATG GCA GAC AGC TTT TCG CT-3′, anti-sense: 5′-GGA ATT CTT AGA TCA TGG CGT GGT TAG CGC TG-3′. The PCR product was subcloned into TA cloning vector pT7-Blue (Takara, Kyoto, Japan) and sequenced. An EcoRI/EcoRI fragment containing the full-length mouse Gal-3 cDNA was inserted into the EcoRI site of the pIRES2-EGFP vector (Clontech Lab, Mountain View, CA). After breaking the Sal1 site between Gal-3 and IRES-GFP, a fragment of this plasmid was blunted and inserted into the EcoR V site of pCAG-loxP-CAT plasmid (a gift from Dr. Araki K., Kumamoto Univ. Kumamoto, Japan). Thus, this plasmid contained the sequence of the CAG promoter, a loxP sequence, CAT gene, SV40 poly (A) signal, a second loxP sequence, mouse Gal-3 cDNA, and IRES-GFP, in that order (Supplementary Fig. 5). A purified Sal1/Not1 fragment of this plasmid was microinjected into fertilized C57BL/6 mouse eggs and 3 independent Flox/Gal-3 mouse strains were generated.

**Chemicals and reagents.** 5-FU (Kyowa Hakko, Tokyo, Japan) was administered to mice intraperitoneally (i.p.) at a dose of 150 mg/kg either as a single dose or once every 7 days two times. Recombinant Ang-1, Thpo and CXCL12 were purchased from R&D Systems (Minneapolis, MN) and diluted with phosphate-buffer saline (PBS). The pharmacological inhibitors PD98059 and LY294002 were purchased from Merck Millipore (Darmstadt, Germany) and diluted with dimethyl sulfoxide (DMSO; Wako, Osaka, Japan). The NF-κB inhibitor BAY-11-7082 (Wako) was reconstituted in DMSO (Wako) as a 10 mM stock solution and diluted with PBS. In animal experiments, BAY-11-7082 was administered i.p. at a dose of 10 mg/kg and mice were sacrificed after 12 h. The Sp1 inhibitor Mithramycin A (MIT) was purchased from Cayman Chemical (Ann Arbor, MI) and diluted with ethanol (Wako). Tolfenamic Acid (TA) was purchased from Wako and mixed with corn oil (Sigma, St. Louis, MO). MIT was administered i.p. once every 2 days, and TA was administered via oral gavage also once every 2 days. Verapamil was purchased from Eisai (Tokyo, Japan) and diluted with sterile water. LPS (Escherichia coli 055:B5) was purchased from FUJIFILM (Osaka, Japan) and diluted with PBS. LPS injections were administered intravenously (i.v.) at a dose of 250 μg/kg.

**Cell culture and transfections.** Cell lines, including Ba/F3 (murine pro-B cell; RCB0805) and OP9 (murine osteoblast; RCB1124) were purchased from the Riken cell bank (Tsukuba, Japan). Ba/F3 cells were cultured in RPMI-1640 medium (Sigma) supplemented with 10% fetal bovine serum (FBS; Sigma) and 1% peni-cillin/streptomycin (Gibco, St. Paul, Brazil) and 0.3 ng/ml IL-3 (Invitrogen, Carlsbad, CA). OP9 cells were cultured in Dulbecco's modified Eagle's medium (DMEM; Sigma) supplemented with 20% FBS, 1% penicillin/streptomycin, and 2 mmol/l L-glutamine (Gibco). Purified LT-HSCs were short-term cultured in RPMI-1640 medium supplemented with 1% FBS, 50 ng/ml mouse stem-cell factor (SCF; PeproTech, Rocky Hill, NJ), and 0.3 ng/ml IL-3 at 37 °C in a humidified atmosphere containing 5% CO$_2$ for 12 h. Ba/F3 cells were transfected with pCMV-mGal-3-IRES2-EGFP containing mouse Gal-3 or with mock vector as a control. Ba/F3 cells were stably transfected using the Lipofectamine 3000 system (Invitrogen) according to the manufacturer's instructions and selected by antibiotic resistance to G418 (Gibco).

**Cre recombinase-inducible Gal-3 overexpression ex vivo.** Cre Recombinase Gesicles (Takara) were added to cultures according to the manufacturer's protocol. Briefly, purified LT-HSCs or LSK cells were plated in 35 mm petri dishes (2 × 10$^5$/dish, FALCON, Durham, NC) containing 1 ml of infection medium [serum-free RPMI-1640 medium plus 6 μg Polybrene (Sigma) and 10 μl Cre Recombinase Gesicles]. After 4 h incubation at 37 °C, medium was changed and cells further cultured in RPMI-1640 supplemented with 1% FBS, 100 ng/ml Thpo, 50 ng/ml SCF, and 0.3 ng/ml IL-3 at 37 °C in a humidified atmosphere containing 5% CO$_2$ in air for 20 h.

**Quantitative real-time PCR.** Total RNA was extracted from cells or tissues using RNeasy-plus mini kits (Qiagen, Hilden, Germany) and was reverse-transcribed using the PrimeScript RT reagent Kit (Takara) according to the manufacturer's protocol. Quantitative real-time PCR analysis was performed with TB Green Premix Ex Taq II (Takara) using the LightCycler 96 System (Roche Diagnostics GmbH, Penzberg, Germany). The baseline and threshold were adjusted according to the manufacturer's instructions. The level of target gene expression was normalized to that of *glyceraldehyde-3-phosphate dehydrogenase* (*GAPDH*) in each sample. We used the primer sets described in Supplementary Table 1.

**Western blotting and Immunoprecipitation.** For western blot analyses, cells (5 × 10$^5$) were washed with ice-cold PBS and lyzed with 50 μl RIPA buffer (Wako). The cells were incubated on ice for 10 min followed by centrifugation at 15,000 rpm for 15 min at 4 °C. Proteins electrophoretically separated using 12.5% SDS-PAGE gels were transferred to polyvinylidene difluoride membranes (GE Healthcare, Chicago, IL) by a wet blotting procedure (140 V, 200 mA, 120 min). The membrane was blocked with 5% skim milk/TBST for 60 min and incubated with the following antibodies: Gal-3/MAC-2 (M3/38; Cedarlane, Ontario, Canada, CL8942AP, 1:500), p21 (EPR3993; Abcam, Eugene, OR, ab109199, 1:200), p57 (EP2525Y; Abcam, ab75974, 1:200), p53 (1C12; Cell Signaling Technology, Danvers, MA, #2524, 1:500), phospho-p53 (Ser15; Cell Signaling Technology, #9284, 1:500), Sp1 (Merck Millipore, 07-645, 1:500), GAPDH (6C5; Merck Millipore, MAB374, 1:2000). Proteins were detected with horseradish-peroxidase-conjugated goat anti-mouse (115-035-003), anti-rat (112-035-003) and anti-rabbit (111-035-003) IgG (Jackson Laboratories, 1:1000) secondary antibodies, and ECL reagents (GE Healthcare). The blots were scanned with an imaging densitometer Amersham Imager 680 system (GE Healthcare). For immunoprecipitation, 500 μl of cell lysate (1 × 10$^6$) was incubated with 5 μl of antibody [rat anti-Gal-3 (M3/38; Cedarlane) or rat anti-IgG (RTK2758; BioLegend, San Diego, CA)] on a rotating device at 4 °C overnight, the 50 μl of Protein-G Sepharose 4 Fast Flow beads (GE Healthcare) were added the following day and incubated at 4 °C for 4 h. The beads were washed six times in cold RIPA buffer (Wako), and supernatant was discarded. Beads were resuspended in 50 μl 2 × SDS-sample buffer (125 mM Tris-HCl, 4% SDS, 10% sucrose, 0.004% bromophenol blue, 10% 2-mercaptoethanol), and heated at 95 °C for 5 min. The samples were fractionated by SDS-PAGE and analyzed by western blotting. The uncropped blots can be found in Supplementary Figs. 9, 10.

**Immunohistochemistry.** For the preparation of BM sections, fresh femoral or tibial bone from C57BL/6 mice was embedded in super cryoembedding medium (Section-lab, Hiroshima, Japan) and frozen in liquid nitrogen. 7–10 μm cryostat sections were generated via the Kawamoto film method. For cell staining, LT-HSCs or cultured cells (1 × 10$^5$) were spun down onto glass slides. BM tissue or cells are fixed for 10 min at room temperature with 4% paraformaldehyde in PBS followed by 2–3 washes with PBS. Tissues and cells were blocked with blocking reagent (5% normal goat serum, 1% BSA and 2% skim milk in PBS) for 1 h at room temperature

and immunostained with primary antibodies. The primary antibodies were as follows: Gal-3/MAC-2 (M3/38; Cedarlane, 1:100), c-Kit/CD117 (R&D Systems, AF1356, 1:100), CD150-PE (TC15-12F12.2; BioLegend, 115904, 1:50), CD150-Alexa Fluor 647 (TC15-12F12.2; BioLegend, 115918, 1:50), Endomucin (V.7C7; eBioscience, San Diego, CA, 14-5851-82, 1:100), Endomucin-PE (V.7C7; eBioscience, 12-5851-82, 1:50), p65 (D14E12; Cell Signaling Technology, #5970, 1:100), p21 (eBiosciences, 14-6715, 1:100), Sp1 (Merck Millipore, 07-645, 1:100) and the following secondary antibodies were used: anti-rat IgG Alexa Fluoe 488 (A-11006), 546 (A-11081), 647 (A-21247; Invitrogen, 1:300), anti-goat IgG Alexa Fluoe 488 (A-11055), 546 (A-11056; Invitrogen, 1:300), and anti-rabbit IgG Alexa Fluoe 546 (A-11010; Invitrogen, 1:300). Cell nuclei were visualized with TO-PRO-3 or DAPI (Invitrogen, 1:1000). For whole mount embryo immunohistochemistry, E9.5 embryos were dissected from decidua, fixed for 4 h at 4 °C with 4% paraformaldehyde in PBS followed by 3 washes with PBS. Embryos were blocked with blocking reagent (5% normal goat serum, 1% BSA, 2% skim milk, and 0.1% Triton-X 100 (Nacalai tesque, Kyoto, Japan) in PBS) for 4 h at 4 °C. The primary antibody, purified rat anti-mouse CD31 (MEC13.3; BD Biosciences, San Jose, CA, 553370, 1:200) was added for overnight at 4 °C. Protein reacting with antibody was visualized with anti-rat-IgG-HRP (Jackson Laboratories, 112-035-003, 1:300). The sections and embryos were examined by conventional microscopy (DM5500 B; Leica, Wetzlar, Germany), confocal microscopy (TCS/SP5; Leica), stereo microscope (M165FC; Leica), and images were acquired with a digital camera (DFC500; Leica). In all analysis, an isotype-matched control Ig was used as a negative control to confirm that the positive signals were not derived from nonspecific background staining. Images were processed using Photoshop CS6 software (Adobe Systems, San Jose, CA) and Image J software (U. S. National Institutes of Health, Bethesda, Maryland). Measurement of the distance between Gal-3+ or Gal-3− HSCs and blood vessels was performed using Volocity software (PerkinElmer, Waltham, MA).

**Flow cytometry and cell sorting**. BM was collected from femora and tibiae. Dispersed BM and fetal liver were drawn through a 23G needle and debris and aggregated cells were removed through a nylon-mesh. Red blood cells were removed from mouse tissues using RBC lysis buffer (Sigma) and then stained with fluorescence- or biotin-conjugated antibodies. Monoclonal antibodies (mAbs) used for this assay were Mouse Lineage Antibody Cocktail (BD Biosciences, 561317, 1:200), Gr-1 (RB6-8C5; BD Biosciences, 11-5931-85, 1:200), Mac-1 (M1/70; BD Biosciences, 553310, 1:200), B220 (RA3-6B2; BD Biosciences, 11-0452-85, 1:200), CD4 (RM4-5; BD Biosciences, 11-0042-85, 1:200), CD8 (53-6.7; BD Biosciences, 11-0081-85, 1:200), CD34 (RAM34; eBioscience, 13-0341, 1:200), CD45 (30-F11; eBioscience, 17-0451-82, 1:200), CD45.1 (A20; BioLegend, 110730, 1:200), Flt3/CD135 (A2F10; BioLegend, 135310, 1:200), Sca-1 (E13-161.7 and D7; BioLegend, 122514 and 108128, 1:200), c-Kit (2B8; BioLegend, 105826, 1:200), CD93 (AA4.1; BioLegend, 136510, 1:200), CD150 (TC15-12F12.2; BioLegend, 115904 and 115926, 1:200), CD48 (HM48-1; BioLegend, 103404 and 103412, 1:200), and phospho-AKT (Ser473; Cell Signaling Technology, #4060, 1:200). All mAbs were purified and conjugated with either Brilliant Violet 421, FITC, PE, APC, APC/Cy7, PE/Cy7, PerCP/Cy5.5, or biotin. Biotinylated antibodies were visualized with PE-conjugated streptavidin (BD Biosciences, 554061, 1:200). In the SP cell procedure, BM cells were resuspended at $1 \times 10^6$ cells/ml and incubated with Hoechst 33342 (5 µg/ml; Sigma) at 37 °C for 90 min. Cells were washed and stained for cell surface markers (anti-Lineage, Sca-1 and c-Kit antibodies). Data acquisition was performed using BD FACS Diva software (BD Biosciences). Analysis was performed using FlowJo software V10 (FlowJo, LLC, San Carlos, CA) and sorted by a FACSAria. Dead cells were excluded by propidium iodide (PI, Sigma) staining or analyses using the two-dimensional profile of the forward versus side scatter. Gating strategy from viable cells are shown in Supplementary Fig. 8.

**Cobblestone area-forming cell (CAFC) assay**. Freshly-isolated BMMNCs ($5 \times 10^5$/well) were seeded onto an 80–90% confluent monolayer of OP9 cells in six-well tissue culture plates (FALCON) in 10% FBS-containing RPMI-1640 medium supplemented with $10^{-5}$ mol/l 2-mercaptoethanol, 20 ng/ml IL-6, 50 U/ml IL-7, 50 ng/ml SCF, and 2 U/ml Epo (all cytokines were from Invitrogen). After 1 week, CAFCs were scored under an inverted microscope (DMi8; Leica).

**Colony-forming unit-cell (CFU-C) assay**. Purified LSK cells from BM or fetal liver ($1 \times 10^3$/dish) were plated in triplicate in 35-mm petri dishes (FALCON) containing 1 ml MethoCult GF M3434 medium (StemCell Technologies Vancouver, BC). After 10 days incubation at 37 °C in 5% $CO_2$ in air, CFU-GM, G, M, GEMM, and BFU-E were counted.

**Colony-forming unit-spleen (CFU-S) assay**. Recipient mice were irradiated with 10 Gy 1 day before transplantation. Whole BM cells ($2 \times 10^5$) from Gal-3+/+ or Gal-3−/− mice were infused via the tail vein into recipient mice. Spleens were isolated 8 or 13 days after transplantation and fixed in Telly's solution. The number of macroscopic spleen colonies was counted and the spleen was weighed.

**Long-term culture-initiating cell (LTC-IC) assay**. Purified LSK cells from Gal-3+/+ or Gal-3−/− mice were co-cultured on OP9 cell layers which had been

inactivated by irradiation (30 Gy) 1 day prior to the experiments. LSK cells ($5 \times 10^3$/well) were cultured with irradiated OP9 cells for 4 weeks in MyeloCult M5300 medium (Stemcell Technologies) supplemented with hydrocortisone (Stemcell Technologies); 50% of the medium was changed weekly. To measure LTC-ICs, the cells were transferred to semisolid MethoCult GF M3434 medium (StemCell Technologies) and cultured for additional 10 days, after which the colonies were counted.

**Cell-cycle analysis**. Ba/F3 or BaF/Gal-3 ($1 \times 10^6$) cells were resuspended and fixed in cold 70% ethanol/ PBS at −20 °C overnight using the PI method. Fixed cells were twice washed with PBS and resuspended in 500 µl PI staining solution [50 µg/ml PI, 0.05% Triton X-100 (Nacalai tesque) and 0.1 mg/ml RNase A (Cell Signaling Technology)] for 20 min at room temperature in the dark. Cell-cycle analysis was performed using FACSCalibur flow cytometry. For the Ki-67 and DAPI labeling method, LT-HSCs from BM or fetal liver were incubated with DAPI and Ki-67 (SolA15, eBioscience, 14-5698-82, 1:200) to determine the cell-cycle profile. The Ki-67 antibody allows for the separation of cells in $G_0$ and $G_1$ stages, and co-staining with DAPI allows for the separation of $S/G_2/M$ cell populations. Cells were analyzed using FACSAria flow cytometry.

**EdU incorporation**. Analysis of EdU incorporation into LT-HSCs from BM was carried out in vivo by injecting mice i.p. with 50 mg/kg of EdU. BM cells were harvested 15–18 h after injection. Purified LT-HSCs were stained for surface antigens using the Click-iT Plus EdU Alexa Fluor 488 Flow Cytometry Assay Kit (C10632; Invitrogen) according to the manufacturer's instructions. Analysis of LT-HSCs (BM or fetal liver) incubated with EdU of in vitro at a final concentration of 10 µM in culture medium for 2 h was by flow cytometry.

**Apoptosis assay**. Apoptotic cells were analyzed by the APC-Annexin V/Dead Cell Apoptosis Kit (V35113; Invitrogen) according to the manufacturer's instructions. Briefly, LT-HSCs from BM or fetal liver were stained with reagents provided in the kit and analyzed by FACSAria flow cytometry. The Sytox Green fluorescence versus Annexin V APC fluorescence dot plot showed resolution of live, apoptotic, and dead cells, which were quantified with the FlowJo V10 software.

**Competitive reconstitution assay**. Five hundred purified LT-HSCs (CD150+ CD48−Flt3−LSK) from Gal-3+/+ or Gal-3−/− (Ly5.2; CD45.2) donors together with $5 \times 10^5$ competitor Ly5.1 (CD45.1) BM cells were transplanted into lethally-irradiated Ly5.1 WT mice (10 Gy) through tail vein injection. After 16 weeks, we assessed chimaerism of donor-derived LT-HSCs (CD45.1−CD150+CD48−Flt3− LSK) and peripheral blood in recipient mice. Then 500 sorted donor-derived LT-HSCs together with new Ly5.1 competitor were transplanted into a second set of lethally-irradiated Ly5.1 WT mice. Subsequent transplantations were performed in the same manner. Percentages of donor-derived LT-HSCs and blood cells were analyzed by FACSAria flow cytometry. Reconstitution of donor myeloid and lymphoid cells was monitored by staining blood cells with antibodies against CD45.1, Mac-1, Gr-1, B220, CD4, and CD8.

**Bone marrow transplantation (BM-T) chimeric mouse model**. Purified LSK cells from Ly5.1 WT mice were transplanted into Gal-3+/+ or Gal3−/− mice (Ly5.2). BM-T was performed using lethally-irradiated (10 Gy) Gal-3+/+ or Gal3−/− mice by intravenous infusion of $5 \times 10^3$ donor-derived LSK cells. Analysis of BM-T chimeric mice was at 16 weeks after transplantation.

**Chromatin immunoprecipitation (ChIP)**. ChIP assays were performed on Ba/F3 or BaF/Gal-3 cells using the SimpleChIP Plus Enzymatic Chromatin IP Kit (Cell Signaling Technology) according to the manufacturer's instructions. Briefly, cells ($2 \times 10^7$) were cross-linked for 10 min in 1% formaldehyde (Wako). Cells were lysed and digested chromatin obtained after Micrococcal Nuclease treatment and sonication. The chromatin was fragmented into a range of sizes from 150 to 900 bp. For the immunoprecipitation (IP), 2% of the chromatin from each treatment condition was stored as the input control and 5 µg of chromatin were used in each IP experiment. Lysates were pre-cleared with Protein-G magnetic beads and incubated at 4 °C overnight with either anti-Gal-3 (M3/38, 5 µl/sample) or anti-histone H3 XP (10 µl/sample), normal anti-rabbit IgG (2 µl/sample) was used as control. After washing the following day, the antibody-protein-DNA complexes were eluted by ChIP elution buffer provided in the kit, and the DNA was purified afterwards. DNA samples were used as a template in PCR, detected by primers for the p21 gene promoter (−2902 to −1). Each ChIP was carried out at least three times with similar results. The primers used are described in Supplementary Table 2.

**RNA Interference**. siRNA specific for mouse Sp1 as well as negative control siRNA were purchased from Sigma and transfected into Ba/F3 or BaF/Gal-3 cells using Amaxa Cell Line Nucleofector Kit V and Nucleofector system (Lonza, Cologne, Germany) according to the manufacturer's instructions. Gene expression changes were analyzed at the protein level by western blotting after 48 h post transfection.

**Statistics and reproducibility**. All data are presented as the means ± standard deviation (S.D.). Statistical analyses were performed using GraphPad Prism 9 (GraphPad Software, Inc., San Diego, CA). Data were analyzed by ANOVA, followed by Tukey-Kramer multiple comparison tests. When only two groups were compared, a two-sided Student's $t$ test was used. A $p$ value < 0.05 was considered statistically significant.

We performed each experiment at least three times with similar results and showed representative data. The level of Sp1, Gal-3, p21, p57, p53, p53-phosphorylation, and GAPDH were checked with western blot analysis. The experiments were replicated at three times with similar results. Representative data are shown in Fig. 6b–d, Supplementary Figs. 4c, g, 6c.

**Reporting summary**. Further information on research design is available in the Nature Research Reporting Summary linked to this article.

## Data availability
The authors declare that all data supporting the findings of this study are available within the article and its Supplementary Information files. Supplementary figs. 1–10 are provided with the paper. Any other data are available from the authors upon reasonable request. Source data are provided with this paper.

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

## Acknowledgements

We thank Ms. Y. Mori, N. Aikawa, and H. Morimoto for administrative assistance. This work was partly supported by Japan Society for the Promotion of Science (JSPS) Grants-in-Aid for Scientific Research (S) (20H05698), (19K16741), the Japan Agency for Medical Research and Development (AMED) under Grant number (20gm5010002s0004, 20cm0106508h0005).

## Author contributions

W.J. and N.T. designed and performed most of the experiments. L.K., H.K., H.N., F.M., Y.H., H.H., and D.Y. supervised assays and performed some experiments. D.H. and F.L. contributed Gal-3$^{-/-}$ mice. W.J. and N.T. analyzed the results and wrote the paper.

## Competing interests

The authors declare no competing interests.
