## [Peer Review File · Nature Communications]

Reviewers' comments:

Reviewer #1 (Remarks to the Author):

Dear Authors,

The manuscript by Jia et al has described that Gal-3 is upregulated by Tie2/Mpl activation to maintain quiescence. The authors have shown 1) Gal-3 regulates cell cycle and differentiation of HSCs using Gal-3-deficient and transgenic mouse model; 2) Ang-1/Tie2 and Thpo/Mpl signal regulates Gal-3 expression and Gal-3 regulates p21 transcription by forming a complex with Sp1. The manuscript is well written and the story is quite interesting with valuable addition to the field. However, the mechanistic insight was drawn by study with pro-B cell line Ba/F3 but not HSC or progenitors from KO/transgenic mice presumably due to limited cell number of HSCs. Some data are not fully convincing and a part of the story, especially p21 protein stabilization by Gal-3, is not supported by provided data. Nevertheless, I believe adding more data from HSCs would make the story more solid. The following concerns should be addressed.

Major comments:

-The statement in page 3 that although these data strongly suggest..... is not entirely supported by already existing papers as the data from the other group have suggested that deficiency of angiopoietin-1 was dispensable for steady state maintenance of HSCs (Zhou BO, Elife 2015). This should be toned down in that the room for discussion remains to be opened.

-Figure 1d: in this figure, the authors claimed Gal-3+c-Kit+ cells are located close to endomucin+ cells. This seems to be true, but Gal-3- cells also seems to be close to endomucin+ cells. Are Gal-3+ cells located significantly closer to endomucin+ cells than Gal-3- cells? Quantification and statistics to compare endomucin positive versus negative Kit+ cells should be done.

-Figure 1e: it would be nicer to see the correlation between expression of Gal3 mRNA and percentage and/or number of G0 cells in the HSCs isolated here post 5FU treatment. Also, any correlation with distance to endomucin+ sinusoid cells upon 5FU?

-Page 7 line5-7: though the authors wrote "Over the time course of BM recovery, we found that the expression of Gal-3 gradually increased again in cells identified as LT-HSCs (CD34-Flk2-LSK) and in CD150+ cells (Supplementary Fig. 1c and Fig. 1e)," the Supplementary Fig.1c doesn't show such a recovery of Gal-3 expression.

-page 9: it was stated that "to eliminate the possibility that Gal-3 deficiency in the stromal cell compartment of the BM...., we analyzed Gal-3 expression in BM stromal cells...." However, only osteoblast and endothelial cells were analyzed afterwards, making it hard to understand the logic. Is endothelial cell considered as stromal cells? If not, are osteoblast cells only stromal cell that exist in BM? The rationale should be explained. To clarify the role of Gal-3 in hematopoietic vs non-hematopoietic cells, one should generate chimeric Gal-3+/+ or -/- mice with Gal-3+/+ or -/- hematopoietic cells reconstituted, and test which combination phenocopies global knock out mice, though Gal-3+/+ or -/- hematopoietic cell transplantation did not show difference in Fig 3h.

-Figure 3d: it is not clear from the story flow if it is necessary to have the schematic image of quiescent (G0) and activated (G1) HSCs in the left panel. If so, it should rather be explained in the text.

Figure 4d: it should have an additional figure separately or in the same which shows the proportion of each differentiated cell type as figure 2g, in order to support the author's claim.

-Page 11 line 12-14: it was mentioned "In later analyses, we have focused on p21 because alteration of this protein is commonly observed in Gal-3-deficient LT-HSCs and Gal-3-overexpressing Ba/F3 cells." The data on p21 protein in Gal-3-deficient LT-HSCs is not there and should be shown.

-Figure 5: Here almost all data are based on the study with Ba/F3 cell line because of limited HSC cell number, but if some data from HSCs are provided, the conclusion would become more solid and convincing. For example, is it possible to show immunostaining (showed in Figure 5j) in HSCs? How about Tie2 and AKT phosphorylation analysis (showed in Figure 5d and 5i) by FACS?

-Supplementary Figure 5: These are quite interesting data in vivo. Some of these data should be replaced with data in main Figure 5.

-Figure 6: the authors conclude Gal-3 regulates p21 protein stability, but no data supporting this conclusion are shown. Although Gal-3-p21 binding is shown, this doesn't necessarily mean Gal-3 regulates p21 protein stability. Because p21 transcription is regulated by Gal-3 (Figure 3e), it would be natural to see low p21 protein expression in Gal3^{-/-} shown in Figure 6a-b

-Figure 6d: p21 bands are not convincing at all. They look unspecific bands. The positive control should be shown.

Minor points

-Figure 5h: Two DMSO control lanes are shown. One would be enough.

-Figure 6b: It is not clear the definition of p21^{high} cells.

-Figure 6d and 6g: IP input lane should be shown.

Reviewer #2 (Remarks to the Author):

In the manuscript entitled "Indispensable role of Galectin-3 in promoting quiescence of hematopoietic stem cells" by Jia et al., the authors show that Gal3 is expressed in HSCs and that Gal-3-positive cells localize in the vascular niche. The authors then delete or overexpress (oe) Gal3 and check the role in HSC quiescence (adult HSCs and fetal liver HSCs respectively). Finally, Jia et al., address the mechanism and show that Gal3 regulates HSC quiescence via p21.

Major concerns:

1) The authors show in Figure 1b the expression of Gal-3 and link the high expression with LT-HSCs. What about the cells that are Gal-3-high in MPP and ST-HSCs? and the low Gal3 in LT-HSCs? These data would argue for heterogeneity. The authors should address this.

2) The authors show in Figure 1d the expression of Gal-3 together with cKit and show that Gal3-kit⁺ cells are expressed close to endomucin⁺ cells. However, this staining looks completely different than Fig. 1b where LT-HSCs are high. Here it seems that the authors are marking another population rather than LT-HSCs. Other markers should be used to determine this discrepancy e.g. CD150.

3) The authors show that Gal3 expression is higher in LT-HSCs compared to other progenitors, however when searching in other databases (e.g. Immgen or http://blood.stemcells.cam.ac.uk/single_cell_atlas.html#RNA_diffplot) the expression looks different –

higher in progenitors and/or not changed. The authors should comment on this discrepancy.

4) The authors should perform CFU secondary plating from Fig 2e to show long-term self-renewal capacities.

5) Fig3c. The authors should use additional markers in their cell cycle profile (now only LSK CD34-). In addition, the cell cycle profile looks technically not convincing. Thus, the authors should i) show dots instead of contour plots and ii) perform cell cycle using an alternative system (e.g. Ki67 + Hoechst or Ki67 + DAPI)

6) What is the phenotype of progenitors in Gal3KO mice?

7) Sup3f. The authors performed the transplants experiments using the CD45.1-2 system. However and according to the gating strategy, the authors are not excluding CD45.1 (potential radio-resistant cells which might vary from mouse to mouse). Thus, the authors should exclude CD45.1 and represent CD45.2 in relation to CD45.1/2.

8) The authors show the effect of 5FU, which is an aggressive chemotherapeutic agent. What about the effect using other milder HSC-proliferation stresses e.g. LPS, pIC?

9) To rule out a niche effect, the authors should support their QPCR findings with reverse chimeras.

11) The authors should perform the 5FU experiments using chimeras to exclude niche effects.

12) The authors should perform proliferation assays.

13) The authors generate a tg -Gal3 oe mouse model, however the expression is only tested in development and not in adulthood. Since the whole study is based on adult HSCs and not fetal liver HSCs, the authors should perform the experiments (e.g. qpcr, CFUs, cell cycle, etc) with an adult inducing mouse model. Alternatively, the authors could perform the experiments using LSK cells instead of Ba/F3 cells (done in Sup fig3a).

Minor comments:

The authors should add references in the introduction. Specially in the first paragraph.

Reviewer #3 (Remarks to the Author):

The manuscript by Jia et al describes the role of Galectin 3 in the maintenance of HSC quiescence. Through the analysis of Galectin 3 deficient mice and Gal3 transgenic mice they concluded that Galectin 3 has an indispensable role in regulating p21 mediated inhibition of cell cycle entry in HSCs. In addition, they attempted to identify the functions of Galactin3 at a molecular level.

While there are no major issues with the technical quality of work presented here, there are several major and serious concerns that were identified with the immunophenotyping strategy, data interpretation and novelty/significance of the presented work.

A few of them are highlighted below;

1. The authors performed their entire study using CD34-Flt3- LSK cells and refer to this fraction as "long-term (LT)-HSCs". This raises a major concern regarding their interpretation of their data, because referring to CD34-Flt3- LSK cells of the BM as LT-HSCs is an "outdated nomenclature" and it has been unequivocally accepted in the field of stem cell biology that only a very minor fraction (~10%) of CD34-Flt3- LSK cells is real LT-HSCs. In fact, there are a number of articles published in the past 15 years discouraged the idea of referring CD34-Flt3- LSK cells as LT-HSCs. The original studies from the Morrison Lab demonstrated that LT-HSCs are CD150+CD48-LSK (Kiel, Cell, 2005 & Oguro, Cell stem cell, 2013) and a refined strategy to identify LT-HSCs based on CD150+CD48-CD34-Flt3-LSK immunophenotyping has been proposed by the Trumpp Laboratory (Wilson, Cell, 2008). More recently, a study from the Passegue lab has demonstrated that LT-HSCs can be identified with CD150+CD48-Flt3-LSK immunophenotype (Pietras, Cell Stem Cell, 2015). In view of the fact that they conducted almost all their studies using CD34-Flt3- LSK cells and that no data has been provided regarding the frequencies of the real LT-HSCs (CD150+CD48-Flt3-LSK), it is very doubtful if any of their findings can be attributed to the true LT-HSCs.
2. Similar to the concern indicated above, the authors referred to CD34+Flt3-LSK cells as ST-HSCs and CD34+Flt3+LSK cells as MPPs. Again, this scheme is not consistent with the current immunophenotype strategy of ST-HSCs and MPPs (Pietras, Cell Stem Cell, 2015).
3. Along the lines mentioned above, the gating strategy for CMPs, GMPs and MEPs are inappropriate. The field has convincingly shown by many groups and different technologies, that true common myeloid progenitors cannot be sorted accurately by Akashi et al. 2000 Nature. The described CMP population is a mixture of pre-GMPs and pre-MEPs that are already restricted to GM or MeK. The authors should adopt the accepted immunophenotyping strategy (Pronk et al. Cell Stem Cell 2007 and Rieger et al. 2008 Brit J Haematol.).
4. The authors refer to Lin-ckit+AA4.1+ cells as "HSC FRACTION IN THE LIVER". This is not true. Even though they cite the paper from the Weissman group, in that original article doesn't claim this fraction as LTHSCs. They simply refer to this fraction as precursors of myeloid/lymphoid progenitors. There are many recent reports available on identifying HSCs from the fetal liver of mice and the authors should characterize them based on this.
5. The authors gate CD45-Ter119-CD31-Sca1- fraction of the bone associated fraction and claim that these are osteoblasts. While it may be true that some of these cells might be osteoblasts, they should have included additional staining such as CD51 to identify osteoblasts, as such the fraction includes all dead cells and many other cells (including fibroblasts) that can be present in the BM niche.
6. Galectin3 has been shown to be ubiquitously expressed in multiple tissues, including heart, the kidney and blood vessels (Endre, 2017). There are a number of cell types that express galectin-3 such as neutrophils, macrophages, and mast cells, and lung, stomach, colon, uterus, and ovary cells (Kim et al., 2007). In view of the fact that Galectin3 has pleiotropic effects on multiple cell types, the relevance and significance of the current study remains unclear, as its role in HSCs is not specific.
7. As clearly pointed out by the authors others, the role of Galectin 3 in the control of cell cycle has been well established. More importantly, previous studies identified a key role for Galectin 3 in the regulation and stability of p21 (Wang, Oncogene, 2013). It is unclear as why the authors conducted extensive studies to recapitulate the already available information.
8. The relevance of Gal3 Transgenic mice is unclear. It would have been useful if the authors had included data on the conditional Galectin mice, to ablate Galectin specifically in HSCs. As of now, all the studies (except BMT) were conducted on total KO mice. Given the fact that Galectin has roles in multiple cell types the specificity and significance of the current studies are questionable. Even though the authors performed BMT experiments, the donor cells were derived from total KOs, the effect that they see in WT recipients could be an impact of "niche defects".
9. The use of Baf/Gal3 cell line (which is a proB cell line) to study molecular mechanisms of "LT-HSCs" is unjustified. The mechanisms of p21 regulation in Baf/Gal3 cell line might be totally different from HSCs. The authors could have shown p21 protein expression in "LT-HSC" by microscopy or by FACS.

10. Galectin 3 has been identified to have a dominant role in inflammation. In view of that fact that inflammation has a critical role in HSC maintenance and functions, it would be necessary that the authors exclude the potential role of inflammation in their HSCs phenotype.

Response to Reviewers' comments:

Reviewer #1 (Remarks to the Author):

Dear Authors,

The manuscript by Jia et al has described that Gal-3 is upregulated by Tie2/Mpl activation to maintain quiescence. The authors have shown 1) Gal-3 regulates cell cycle and differentiation of HSCs using Gal-3-deficient and transgenic mouse model; 2) Ang-1/Tie2 and Thpo/Mpl signal regulates Gal-3 expression and Gal-3 regulates p21 transcription by forming a complex with Sp1. The manuscript is well written and the story is quite interesting with valuable addition to the field. However, the mechanistic insight was drawn by study with pro-B cell line Ba/F3 but not HSC or progenitors from KO/transgenic mice presumably due to limited cell number of HSCs. Some data are not fully convincing and a part of the story, especially p21 protein stabilization by Gal-3, is not supported by provided data. Nevertheless, I believe adding more data from HSCs would make the story more solid. The following concerns should be addressed.

Response to Reviewer #1

The authors thank Reviewer 1 for their comments and valuable suggestions to improve our manuscript. Please see our point-by-point responses to the comments below.

Major comments:

Comment 1-1:

-The statement in page 3 that although these data strongly suggest..... is not entirely supported by already existing papers as the data from the other group have suggested that deficiency of angiopoietin-1 was dispensable for steady state maintenance of HSCs (Zhou BO, Elife 2015). This should be toned down in that the room for discussion remains to be opened.

Our response to Comment 1-1:

As the reviewer suggested, the statement is open to misunderstanding. Therefore, we have rewritten this sentence as follows: Of the many cytokine receptors expressed by HSCs, Tie2 and Mpl have been reported to be involved in maintaining the quiescent state. Although it has been argued that Angiopoietin-1 (Ang-1)/Tie2 has a non-essential role, enhanced signaling via Tie2 by Ang-1, and Mpl signaling by Thrombopoietin (Thpo), seem to induce HSC quiescence by activating cell adhesion molecules such as β 1-integrin and N-cadherin. However, the mechanisms by which Tie2 and Mpl maintain quiescence have not been fully elucidated. (lines 10-16 page 3)

Comment 1-2:

-Figure 1d: in this figure, the authors claimed Gal-3+c-Kit+ cells are located close to endomucin+ cells. This seems to be true, but Gal-3- cells also seems to be close to endomucin+ cells. Are Gal-3+ cells located significantly closer to endomucin+ cells than Gal-3- cells? Quantification and statistics to compare endomucin positive versus negative Kit+ cells should be done.

Our response to Comment 1-2:

In accordance with this suggestion, we re-stained the bone marrow (BM) with antibodies against Gal-3, CD150, and Endomucin and performed quantitative analysis of the distance between Gal-3⁺HSC or Gal-3⁻HSC and BM blood vessels. We found that Gal-3⁺ HSCs located closer to the blood vessels than Gal-3⁻HSCs. Gal-3⁺ HSCs were mostly present in an area within 30 μ m from the blood vessel, revealing that the majority of Gal-3⁺ HSCs were associated with the vascular niche. We have added this information to the Results section (line 12 page 6) and Fig. 1 d, e.

Comment 1-3:

-Figure 1e: it would be nicer to see the correlation between expression of Gal3 mRNA and percentage and/or number of G₀ cells in the HSCs isolated here post 5FU treatment. Also, any correlation with distance to endomucin+ sinusoid cells upon 5FU?

Our response to Comment 1-3:

We analyzed the relationship between Gal-3 expression levels and the proportion of LT-HSC (phenotypically) in G₀ phase. We found that during 5-FU-mediated BM reconstitution, Gal-3 expression in LT-HSC also changed. Four days after 5-FU injection, the lowest amount of LT-HSC was detected in G₀ phase and Gal-3 expression was significantly attenuated in LT-HSC. From 6 days after 5-FU injection, the proportion of LT-HSC in the G₀ phase gradually increased and the level of expression of Gal-3 gradually recovered. These results suggest that the change of Gal-3 expression in LT-HSC plays a role in the maintenance of HSC quiescence during BM reconstitution. We have added this information to the Results section (lines 13-18 page 6, lines 1-8 page 7) and Supplementary Fig. 1 d-f.

In addition, we also performed quantitative analysis of the distance between Gal-3⁺HSC or Gal-3⁻HSC and BM blood vessels at different time points after 5-FU injection. We found that compared with Gal-3⁻HSCs, Gal-3⁺HSCs interact more with the vascular niche. Eight days after 5-FU injection, the proportions of Gal-3⁺HSC significantly increased in the area within 15 μ m distance from the blood vessel, relative to 4 days after 5-FU injection. However, there was no significant change in the Gal-3⁻HSCs. We have added this information to the Results section (lines 8-17 page 7) and Fig. 1 f, g.

Comment 1-4:

-Page 7 line5-7: though the authors wrote “Over the time course of BM recovery, we found that the expression of Gal-3 gradually increased again in cells identified as LT-HSCs (CD34-Flk2-LSK) and in CD150+ cells (Supplementary Fig. 1c and Fig. 1e),” the Supplementary Fig.1c doesn't show such a recovery of Gal-3 expression.

Our response to Comment 1-4:

In accordance with this suggestion, we modified Supplementary Fig. 1 c and Fig. 1 e, and added the immunostaining results of Gal-3 and CD150 or c-Kit in the BM sections from WT mice at 6 or 8 days after 5-

FU injection. We have added this information to the Results section (lines 13-18 page 6, lines 1-8 page 7) and Supplementary Fig. 1 d, e.

Comment 1-5:

-page 9: it was stated that “to eliminate the possibility that Gal-3 deficiency in the stromal cell compartment of the BM...., we analyzed Gal-3 expression in BM stromal cells....” However, only osteoblast and endothelial cells were analyzed afterwards, making it hard to understand the logic. Is endothelial cell considered as stromal cells? If not, are osteoblast cells only stromal cell that exist in BM? The rationale should be explained. To clarify the role of Gal-3 in hematopoietic vs non-hematopoietic cells, one should generate chimeric Gal-3^{+/+} or Gal-3^{-/-} mice with Gal-3^{+/+} or Gal-3^{-/-} hematopoietic cells reconstituted, and test which combination phenocopies global knock out mice, though Gal-3^{+/+} or Gal-3^{-/-} hematopoietic cell transplantation did not show difference in Fig 3h.

Our response to Comment 1-5:

Thank you for your constructive criticisms. This is also an important question that needs to be answered. To eliminate the possibility that Gal-3 deficiency in the niche (environmental) cell component of the BM, rather than in the HSCs themselves, caused the accelerated differentiation phenotype, we established a BM-transplantation (BM-T) chimeric mouse model. Purified LSK cells from Ly5.1 WT mice were transplanted into lethally-irradiated Gal-3^{+/+} or Gal-3^{-/-} mice (Ly5.2). Sixteen weeks after BM-T we could not distinguish any differences between Gal-3^{+/+} and Gal-3^{-/-} recipient mice in body weight, spleen weight, and the proportion and absolute number of spleen-LSK cells, BM-LSK cells or BM-LT-HSCs. During the process of BM reconstitution (5-FU), Gal-3 deletion in BM niche cells did not affect the HSC reconstitution ability, i.e., the frequency and the absolute number of donor-derived HSCs showed a similar pattern between Gal-3^{+/+} and Gal-3^{-/-} recipient mice. We have added this information to the Results section (lines 12-18 page 10, lines 1-10 page 11) and Supplementary Fig. 3.

Comment 1-6:

-Figure 3d: it is not clear from the story flow if it is necessary to have the schematic image of quiescent (G0) and activated (G1) HSCs in the left panel. If so, it should rather be explained in the text.

Our response to Comment 1-6:

We deleted the results (including schema) shown in Fig. 3 d, added new experimental results and made a new Fig. 3.

Comment 1-7:

-Figure 4d: it should have an additional figure separately or in the same which shows the proportion of each differentiated cell type as figure 2g, in order to support the author's claim.

Our response to Comment 1-7:

In accordance with this suggestion, we show the proportion of each differentiated cell type in Fig. 4 d, and added this information to the Results section (lines 9-10 page 17).

Comment 1-8:

-Page 11 line 12-14: it was mentioned “In later analyses, we have focused on p21 because alteration of this protein is commonly observed in Gal-3-deficient LT-HSCs and Gal-3-overexpressing Ba/F3 cells.” The data on p21 protein in Gal-3-deficient LT-HSCs is not there and should be shown.

Our response to Comment 1-8:

In accordance with this suggestion, we further analyzed p21 protein expression levels in Gal-3^{+/+} and Gal-3^{-/-} BM-derived LT-HSCs using immunofluorescence staining. We observed that levels of p21 protein were significantly lower in the Gal-3^{-/-} LT-HSCs than Gal-3^{+/+} LT-HSCs. We have added this information to the Results section (lines 5-8 page 13) and Fig. 3 d, e.

Comment 1-9:

-Figure 5: Here almost all data are based on the study with Ba/F3 cell line because of limited HSC cell number, but if some data from HSCs are provided, the conclusion would become more solid and convincing. For example, is it possible to show immunostaining (showed in Figure 5j) in HSCs? How about Tie2 and AKT phosphorylation analysis (showed in Figure 5d and 5i) by FACS?

Our response to Comment 1-9:

Thank you for your suggestion. We used data from LT-HSCs derived from WT mouse BM to replace the experimental results using BaF3 cells. This does increase the reliability of the results. We have added this information to the Results section (lines 2-14 page 20) and Fig. 5 d-i.

Comment 1-10:

-Supplementary Figure 5: These are quite interesting data in vivo. Some of these data should be replaced with data in main Figure 5.

Our response to Comment 1-10:

Thank you for your suggestion. We have moved the data of Supplementary Fig. 5 to Fig. 5 and explained this (lines 14-18 page 20, lines 1-6 page 21) and Fig. 5 j-o.

Comment 1-11:

-Figure 6: the authors conclude Gal-3 regulates p21 protein stability, but no data supporting this conclusion are shown. Although Gal-3-p21 binding is shown, this doesn't necessarily mean Gal-3 regulates p21 protein

stability. Because p21 transcription is regulated by Gal-3 (Figure 3e), it would be natural to see low p21 protein expression in Gal3^{-/-} shown in Figure 6a-b

Our response to Comment 1-11:

Thank you for your constructive criticisms. It is indeed a challenge to show regulation of stability of p21 by Gal-3 which has been suggested in other papers. Although we overexpressed Gal-3 in BaF3 cells and proved that Gal-3 protein interacted with p21 protein, we could not accurately determine how Gal-3 regulates p21 protein stability in LT-HSCs. Therefore, with regret, we decided to delete this part.

Comment 1-12:

-Figure 6d: p21 bands are not convincing at all. They look unspecific bands. The positive control should be shown

Our response to Comment 1-12:

Depending on your comment (No. 11), we deleted the data showing p21 in the original Fig. 6 d.

Minor points

1. -Figure 5h: Two DMSO control lanes are shown. One would be enough.

We now use LT-HSC analysis to replace the results using BaF3 cells. Therefore, we deleted the original Fig. 5 h.

2. -Figure 6b: It is not clear the definition of p21^{high} cells.

We used Image J to quantify the fluorescence intensity of p21, and added the results in Fig. 3 d and e.

3. -Figure 6d and 6g: IP input lane should be shown.

Several IP data were deleted but we added the Input lane to the results of the remaining immunoprecipitation data (Fig. 6 c).

Reviewer #2 (Remarks to the Author):

In the manuscript entitled “Indispensable role of Galectin-3 in promoting quiescence of hematopoietic stem cells” by Jia et al., the authors show that Gal3 is expressed in HSCs and that Gal-3-positive cells localize in the vascular niche. The authors then delete or overexpress (oe) Gal3 and check the role in HSC quiescence (adult HSCs and fetal liver HSCs respectively). Finally, Jia et al., address the mechanism and show that Gal3 regulates HSC quiescence via p21.

Response to Reviewer #2

The authors thank Reviewer 2 for their comments and valuable suggestions to improve our manuscript. Please see our point-by-point responses to the comments below.

Major comments:

Comment 2-1:

1) The authors show in Figure 1b the expression of Gal-3 and link the high expression with LT-HSCs. What about the cells that are Gal-3-high in MPP and ST-HSCs? and the low Gal3 in LT-HSCs? These data would argue for heterogeneity. The authors should address this.

Our response to Comment 2-1:

Thank you for your suggestion. We used Image J to quantify Gal-3 protein expression in the nuclei of different cell types (LT-HSC, ST-HSC and MPP2-4). Results showed that the expression of Gal-3 was mainly concentrated in the nucleus in the case of LT-HSCs. These results relate to our analysis showing that Gal-3 regulates the transcription of p21 and promotes quiescence in LT-HSCs. We have now added the results of the heterogeneity of Gal-3 expression (lines 4-6 page 6) and Fig. 1 b, c. We added a discussion of the heterogeneity of LT-HSCs according to their expression of Gal-3 (lines 4-14 page 25).

Comment 2-2:

2) The authors show in Figure 1d the expression of Gal-3 together with cKit and show that Gal3-kit⁺ cells are expressed close to endomucin⁺ cells. However, this staining looks completely different than Fig. 1b where LT-HSCs are high. Here it seems that the authors are marking another population rather than LT-HSCs. Other markers should be used to determine this discrepancy e.g. CD150.

Our response to Comment 2-2:

In accordance with this suggestion, we applied immunofluorescence staining to visualize Gal-3, CD150 and Endomucin in BM sections from Gal-3^{+/+} and Gal-3^{-/-} mice. Moreover, we performed quantitative analyses of the distance between Gal3⁺ HSC and blood vessels. We found that compared with Gal3⁻HSC, Gal3⁺HSCs locate closer to the blood vessels. We added this information to the Results section (line 12 page 6) and Fig. 1 d, e.

Comment 2-3:

3) The authors show that Gal3 expression is higher in LT-HSCs compared to other progenitors, however when searching in other databases (e.g. Immgen or http://blood.stemcells.cam.ac.uk/single_cell_atlas.html#RNA_diffplot) the expression looks different – higher in progenitors and/or not changed. The authors should comment on this discrepancy.

Our response to Comment 2-3:

Thank you for your suggestion. We also noticed that point. Therefore, we used several different methods to purify LT-HSC, i.e., SP^{low}LSK, CD34-Flt3-LSK and CD150⁺CD48-Flt3-LSK, to confirm our results. Other investigators have used CD34-Flt3-LSK methods to isolate HSCs. CD34 staining could not separate two

obvious cell populations (ST-HSCs or MPPs) and this must affect the results of Gal-3 expression analyses. After several trials using different markers, we isolated LT-HSCs as CD150⁺CD48⁻Flt3⁻LSK, although this was because of a suggestion by Reviewer 3. Now, we show data on Gal-3 expression in several fractions of cells (Fig. 1 a).

Comment 2-4:

4) The authors should perform CFU secondary plating from Fig 2e to show long-term self-renewal capacities.

Our response to Comment 2-4:

In accordance with this suggestion, we performed LTC-IC assays to evaluate the effect of Gal-3 deficiency on self-renewal ability of LT-HSCs. We have added this information to the Results section (lines 8-11 page 10) and Fig. 2 i.

Comment 2-5:

5) Fig3c. The authors should use additional markers in their cell cycle profile (now only LSK CD34-). In addition, the cell cycle profile looks technically not convincing. Thus, the authors should i) show dots instead of contour plots and ii) perform cell cycle using an alternative system (e.g. Ki67 + Hoechst or Ki67 + DAPI)

Our response to Comment 2-5:

In accordance with this suggestion, we performed Ki-67 and DAPI double staining to analyze the cell cycle of BM derived- or fetal liver derived-LT-HSCs (CD150⁺CD48⁻LSK). We have added this information to the Results section (lines 2-5 page 12, lines 14-17 page 17, line 18 page 18, lines 1-2 page 19), Fig. 3 a and Fig. 4 e, k.

Comment 2-6:

6) What is the phenotype of progenitors in Gal3KO mice?

Our response to Comment 2-6:

We performed CFU-S assays to investigate short-term hematopoiesis of progenitors (CFU-S₈) and the self-renewal ability of HSCs (CFU-S₁₃). We found that more colonies were formed on spleen after 8 days when Gal-3^{-/-} BM cells were injected than when WT BM cells were. However, 13 days after BM-T, the number of spleen colonies sharply decreased in mice injected with Gal-3^{-/-} BM cells and was then lower compared with WT counterparts. This suggests that HSCs derived from Gal-3^{-/-} mice have lower self-renewal ability. We have added this information to the Results section (lines 2-7 page 10) and Fig. 2 h.

Comment 2-7:

7) Sup3f. The authors performed the transplants experiments using the CD45.1-2 system. However and according to the gating strategy, the authors are not excluding CD45.1 (potential radio-resistant cells which

might vary from mouse to mouse). Thus, the authors should exclude CD45.1 and represent CD45.2 in relation to CD45.1/2.

Our response to Comment 2-7:

In accordance with this suggestion, we estimated CD45.2 cells in relation to CD45.1/2 and therefore, modified Fig. 3 i and Supplementary Fig. 4 h, to further clarify the proportions of donor-derived or competitor-derived cells. Regarding potential radiation-resistant cells in the recipient mice, all BM-T experimental models in this study used lethal radiation (10 Gy), and as we show in Supplementary Fig. 3 m and n, the fraction of potentially radiation-resistant cells is below 3% in the recipient mice.

Comment 2-8:

8) The authors show the effect of 5FU, which is an aggressive chemotherapeutic agent. What about the effect using other milder HSC-proliferation stresses e.g. LPS, pIC?

Our response to Comment 2-8:

We conducted LPS i.v. administration experiments in Gal-3^{+/+} and Gal-3^{-/-} mice. We found that LPS induced the proliferation of quiescent BM LT-HSCs in both Gal-3^{+/+} and Gal-3^{-/-} mice, but this returned to the original level after 72 hrs. Initially, there was a difference between LT-HSC proportions and numbers in Gal-3^{+/+} and Gal-3^{-/-} mice (Fig. 2 c and d), so we evaluated their comparative fold-change. We found no significant differences between Gal-3^{+/+} and Gal-3^{-/-} mice. We have added this information to the Discussion section (lines 13-18 page 26, lines 1-6, page 27) and Supplementary Fig. 7 a, b.

Comment 2-9:

9) To rule out a niche effect, the authors should support their QPCR findings with reverse chimeras.

Our response to Comment 2-9:

Thank you for your constructive criticisms. This is almost same comment as that of Reviewer 1 (No. 5). To eliminate the possibility that Gal-3 deficiency in the niche cell component of the BM, rather than in the HSCs themselves, caused the accelerated differentiation phenotype, we established a BM-T chimeric mouse model. Purified LSK cells from Ly5.1 WT mice were transplanted into lethally-irradiated Gal-3^{+/+} or Gal-3^{-/-} mice (Ly5.2) and 16 weeks after BM-T, we could not see any differences between the two in body weight, spleen weight, or the proportion and amount number of spleen-LSK cells, BM-LSK cells or BM-LT-HSCs. We have added this information to the Results section (lines 12-18 page 10, lines 1-10 page 11) and Supplementary Fig. 3.

Comment 2-10:

10) The authors should perform the 5FU experiments using chimeras to exclude niche effects.

Our response to Comment 2-10:

During the process of BM reconstitution after 5-FU, Gal-3 deletion in BM niche cells did not affect HSC reconstitution ability, i.e., the frequency and the absolute number of donor-derived HSCs showed a similar pattern between Gal-3^{+/+} and Gal-3^{-/-} recipient mice. This suggests that HSC dysfunction in Gal-3^{-/-} mice occurs in an HSC-intrinsic manner and that Gal-3 deficiency in the BM microenvironment (i.e. the niche cells) does not affect HSC function. We have added this information to the Results section (lines 4-8 page 11) and Supplementary Fig. 3 m-o.

Comment 2-11:

11) The authors should perform proliferation assays.

Our response to Comment 2-11:

In accordance with this suggestion, we performed an *in vivo* and an *in vitro* EdU incorporation assay, to compare the proliferation of BM LT-HSCs of Gal-3^{+/+} or Gal-3^{-/-} mice (*in vivo*), and the proliferation of BM or fetal liver LT-HSCs of control or Flox/Gal-3 mice (*in vitro*). We found that Gal-3 deficiency resulted in LT-HSC proliferation, and reciprocally, Gal3-overexpression suppressed their proliferation. We have added this information to the Results section (lines 5-8 page 12, lines 2-4 page 19, lines 17-18 page 17, lines 1-2 page 18), Fig. 3 b, Fig. 4 l and Supplementary Fig. 5 f.

Comment 2-12:

12) The authors generate a tg -Gal3 oe mouse model, however the expression is only tested in development and not in adulthood. Since the whole study is based on adult HSCs and not fetal liver HSCs, the authors should perform the experiments (e.g. qPCR, CFUs, cell cycle, etc) with an adult inducing mouse model. Alternatively, the authors could perform the experiments using LSK cells instead of Ba/F3 cells (done in Sup fig3a).

Our response to Comment 2-12:

In accordance with this suggestion, as shown in Fig. 4 g, we generated a model with Cre Recombinase Gesicles, to overexpress Gal-3 in adult BM LT-HSCs in *in vitro* cultures. This model had higher recombination efficiency *in vitro*, and Gal-3 overexpression in BM adult LT-HSCs or LSK cells from Flox/Gal-3 mice was induced (Fig. 4 h). We used this model to analyze the effect of Gal-3 overexpression in LT-HSCs. We found that colony forming ability was decreased by the overexpression of Gal-3 in adult LSK cells and Gal-3 overexpression in LT-HSC induced cell cycle arrest in the G₀ phase and inhibited proliferation. Gal-3 overexpression induced p21 expression significantly. We have added this information to the Results section (lines 5-18 page 18, lines 1-7 page 19) and Fig. 4 g-m.

Minor points

1. The authors should add references in the introduction. Specially in the first paragraph.

Thank you for the suggestion. We already added the references in the Introduction section (Reference 1-5).

Reviewer #3 (Remarks to the Author):

The manuscript by Jia et al describes the role of Galectin 3 in the maintenance of HSC quiescence. Through the analysis of Galectin 3 deficient mice and Gal3 transgenic mice they concluded that Galectin 3 has an indispensable role in regulating p21 mediated inhibition of cell cycle entry in HSCs. In addition, they attempted to identify the functions of Galactin3 at a molecular level.

While there are no major issues with the technical quality of work presented here, there are several major and serious concerns that were identified with the immunophenotyping strategy, data interpretation and novelty/significance of the presented work.

A few of them are highlighted below;

Response to Reviewer #3

The authors thank Reviewer 3 for their comments and valuable suggestions to improve our manuscript. Please see our point-by-point responses to the comments below.

Major comments:

Comment 3-1:

1. The authors performed their entire study using CD34-Flt3- LSK cells and refer to this fraction as “long-term (LT)-HSCs”. This raises a major concern regarding their interpretation of their data, because referring to CD34-Flt3- LSK cells of the BM as LT-HSCs is an “outdated nomenclature” and it has been unequivocally accepted in the field of stem cell biology that only a very minor fraction (~10%) of CD34-Flt3- LSK cells is real LT-HSCs. In fact, there are a number of articles published in the past 15 years discouraged the idea of referring CD34-Flt3- LSK cells as LT-HSCs. The original studies from the Morrison Lab demonstrated that LT-HSCs are CD150⁺CD48⁻LSK (Kiel, Cell, 2005 & Oguro, Cell stem cell, 2013) and a refined strategy to identify LT-HSCs based on CD150⁺CD48⁻CD34-Flt3-LSK immunophenotyping has been proposed by the Trumpp Laboratory (Wilson, Cell, 2008). More recently, a study from the Passegue lab has demonstrated that LT-HSCs can be identified with CD150⁺CD48⁻Flt3-LSK immunophenotype (Pietras, Cell Stem Cell, 2015). In view of the fact that they conducted almost all their studies using CD34-Flt3- LSK cells and that no data has been provided regarding the frequencies of the real LT-HSCs (CD150⁺CD48⁻Flt3-LSK), it is very doubtful if any of their findings can be attributed to the true LT-HSCs.

Our response to Comment 3-1:

In accordance with this suggestion, as we show in Fig. 1 a, we used CD150⁺CD48⁻Flt3-LSK as LT-HSCs. -----

Comment 3-2:

2. Similar to the concern indicated above, the authors referred to CD34⁺Flt3⁻LSK cells as ST-HSCs and CD34⁺Flt3⁺LSK cells as MPPs. Again, this scheme is not consistent with the current immunophenotype strategy of ST-HSCs and MPPs (Pietras, Cell Stem Cell, 2015).

Our response to Comment 3-2:

In accordance with this suggestion, as we show in Fig. 1 a, we used CD150⁻CD48⁻Flt3⁻LSK as the marker of ST-HSCs, CD150⁺CD48⁺Flt3⁻LSK as the marker of MPP2, CD150⁻CD48⁺Flt3⁻LSK as the marker of MPP3, and CD150⁻CD48⁺Flt3⁺LSK as the marker of MPP4.

Comment 3-3:

3. Along the lines mentioned above, the gating strategy for CMPs, GMPs and MEPs are inappropriate. The field has convincingly shown by many groups and different technologies, that true common myeloid progenitors cannot be sorted accurately by Akashi et al. 2000 Nature. The described CMP population is a mixture of pre-GMPs and pre-MEPs that are already restricted to GM or MekkE. The authors should adopt the accepted immunophenotyping strategy (Pronk et al. Cell Stem Cell 2007 and Rieger et al. 2008 Brit J Haematol.).

Our response to Comment 3-3:

We found that Gal-3 was highly expressed in LT-HSCs relative to ST-HSCs or MPP2-4 cells. Therefore, we have omitted the comparison of Gal-3 expression in LT-HSCs, CLPs, CMPs, GMPs and MEPs. The results were deleted from the main text.

Comment 3-4:

4. The authors refer to Lin⁻ckit⁺AA4.1⁺ cells as “HSC FRACTION IN THE LIVER”. This is not true. Even though they cite the paper from the Weissman group, in that original article doesn’t claim this fraction as LTHSCs. They simply refer to this fraction as precursors of myeloid/lymphoid progenitors. There are many recent reports available on identifying HSCs from the fetal liver of mice and the authors should characterize them based on this.

Our response to Comment 3-4:

Thank you for your constructive criticisms. The fetal site at which the first LT-HSCs emerge is still debated. Nevertheless, most investigators accept that LT-HSC appear in fetal liver at embryonic day 11.5. In order to highly purify LT-HSCs from E12.5 fetal liver, we opted to utilize the CD150⁺CD48⁻LSK as the selection marker. Furthermore, at early stages (E10.5- E12.5), to purify HSC/progenitors in fetal liver, we applied Lineage⁻c-Kit⁺AA4.1⁺ as selection markers.

Comment 3-5:

5. The authors gate CD45-Ter119-CD31-Sca1- fraction of the bone associated fraction and claim that these are osteoblasts. While it may be true that some of these cells might be osteoblasts, they should have included additional staining such as CD51 to identify osteoblasts, as such the fraction includes all dead cells and many other cells (including fibroblasts) that can be present in the BM niche.

Our response to Comment 3-5:

Replacing the cell surface markers of BM niche cells may not completely rule out the impact of Gal-3 deficiency in LT-HSCs, so we generated a BM-T chimeric mouse model instead of discussing the phenotype of niche cells for HSCs. This was because we would like to show the importance of intrinsic Gal-3 in HSCs. Purified LSK cells from Ly5.1 WT mice were transplanted into lethally-irradiated Gal-3^{+/+} or Gal-3^{-/-} mice (Ly5.2) and at 16 weeks after BM-T, we found no differences between the two groups in body weight, spleen weight, and the proportion and amount number of spleen-LSK cells, BM-LSK cells and BM-LT-HSCs. During the process of BM reconstitution after 5-FU treatment, Gal-3 deletion in the BM niche cells did not affect HSC reconstitution ability and the frequency and the absolute number of donor-derived HSCs showed a similar pattern in both Gal-3^{+/+} and Gal-3^{-/-} recipient mice. This suggests that HSC dysfunction in Gal-3^{-/-} mice causes the changes to HSC, and that Gal-3 deficiency in the BM microenvironment (niche cells) does not affect HSC function. We have added this information to the Results section (lines 12-18 page 10, lines 1-10 page 11) and Supplementary Fig. 3.

Comment 3-6:

6. Galectin3 has been shown to be ubiquitously expressed in multiple tissues, including heart, the kidney and blood vessels (Endre,2017). There are a number of cell types that express galectin-3 such as neutrophils, macrophages, and mast cells, and lung, stomach, colon, uterus, and ovary cells (Kim et al., 2007). In view of the fact that Galactin3 has pleiotropic effects on multiple cell types, the relevance and significance of the current study remains unclear, as its role in HSCs is not specific.

Our response to Comment 3-6:

As set out in our response to your comment No. 6, on the basis of the results from the BM-chimeras, we strongly suggest that intrinsic Gal-3 in HSCs is critical for cell cycle control. On the other hand, as the reviewer suggested, Gal-3 expression is ubiquitous over a wide range of cell types. In our paper, therefore, we focused on the hematopoietic lineage, especially HSCs, in order to analyze the function of Gal-3 in hematopoiesis. It is possible that conditional KO of Gal-3 in hematopoietic cells may more strongly suggest the function of Gal-3 in HSCs; however, long-term observation in a serial BM HSC transplantation model is useful to elucidate the HSC-specific function of Gal-3.

Comment 3-7:

7. As clearly pointed out by the authors others, the role of Galectin 3 in the control of cell cycle has been well established. More importantly, previous studies identified a key role for Galectin 3 in the regulation and

stability of p21 (Wang, Oncogene,2013). It is unclear as why the authors conducted extensive studies to recapitulate the already available information.

Our response to Comment 3-7:

In this work, we clarify the function and mechanism of action of Gal-3 in HSCs with the aim of understanding the maintenance of HSC quiescence in the BM niche. Although the regulation of p21 by Gal-3 has been mentioned in cancer research as suggested by the reviewer, there remain many uncertainties concerning the mechanisms responsible. For example, how Gal-3 regulates p21 transcription in cancers has not been elucidated. Gene expression in physiological and pathological conditions is different and it is not known how Gal-3 expression affects p21 in HSCs. Therefore, we believe that our findings are novel. Through this research, we hope to illustrate the regulatory effect of Gal-3 on HSCs via its influence on the cell cycle to maintain their undifferentiated state in vitro. If the reviewer thinks that analysis of p21 in HSC relating to Gal-3 is not necessary in our paper, we will delete these results or move them to Supplementary Results.

Comment 3-8:

8. The relevance of Gal3 Transgenic mice is unclear. It would have been useful if the authors had included data on the conditional Galectin mice, to ablate Galectin specifically in HSCs. As of now, all the studies (except BMT) were conducted on total KO mice. Given the fact that Galectin has roles in multiple cell types the specificity and significance of the current studies are questionable. Even though the authors performed BMT experiments, the donor cells were derived from total KOs, the effect that they see in WT recipients could be an impact of “niche defects”.

Our response to Comment 3-8:

Thank you for the accurate suggestion. Gal-3 is highly expressed in a variety of different cells, and it is necessary to distinguish its function in LT-HSCs. However, we regret that we did not create and analyze conditional Gal-3 KO mice due to time considerations and other constraints. We would like to proceed with that in future Gal-3 studies. However, as described in the response to the reviewer comments (No. 6), we believe that long-term observation in the serial BM HSC transplantation model is useful to elucidate the HSC-specific function of Gal-3.

Comment 3-9:

9. The use of Baf/Gal3 cell line (which is a proB cell line) to study molecular mechanisms of “LT-HSCs” is unjustified. The mechanisms of p21 regulation in Baf/Gal3 cell line might be totally different from HSCs. The authors could have shown p21 protein expression in “LT-HSC” by microscopy or by FACS.

Our response to Comment 3-9:

Thank you for your suggestion. We have now replaced the results from BaF3 cells with data on LT-HSCs derived from WT mice BM. We have added this information to the Results section (lines 2-18 page 20, lines 1-6 page 21) and Fig. 5 d-o.

Comment 3-10:

10. Galectin 3 has been identified to have a dominant role in inflammation. In view of that fact that inflammation has a critical role in HSC maintenance and functions, it would be necessary that the authors exclude the potential role of inflammation in their HSCs phenotype.

Our response to Comment 3-10:

In accordance with this suggestion, we conducted LPS i.v. administration experiments in Gal-3^{+/+} and Gal-3^{-/-} mice. We found that LPS induced the proliferation of BM quiescent LT-HSCs in both Gal-3^{+/+} and Gal-3^{-/-} mice, but this returned to the original level after 72 hrs. Initially, there was a difference between LT-HSC proportions and numbers in Gal-3^{+/+} and Gal-3^{-/-} mice (Fig. 2 c and d), so we evaluated their comparative fold-change. We found no significant differences between Gal-3^{+/+} and Gal-3^{-/-} mice. We have added this information to the Discussion section (lines 13-18 page 26, lines 1-6, page 27) and Supplementary Fig. 7 a, b.

REVIEWERS' COMMENTS

Reviewer #1 (Remarks to the Author):

The authors have appropriately addressed most of my comments.
I have only one minor point.

-Figure 6c: WB images blotted with Gal-3 antibody are missing. Please show them.

Reviewer #2 (Remarks to the Author):

The authors have addressed all of my concerns in the revised version of the manuscript.

Reviewer #3 (Remarks to the Author):

The Authors made a sincere attempt to address most of the concerns raised by this reviewer. The quality of this manuscript is undoubtedly improved and now may be published in this journal.

Response to Reviewers' comments:

Reviewer #1 (Remarks to the Author):

The authors have appropriately addressed most of my comments.

I have only one minor point.

-Figure 6c: WB images blotted with Gal-3 antibody are missing. Please show them.

Our response to this comment:

We have added this information to Figure 6c.